# Scaling Unsupervised Multi-Source Federated Domain Adaptation through Group-Wise Discrepancy Minimization

**Larissa Reichart** [* 1]   **Cem Ata Baykara** [* 1]   **Ali Burak Ünal** [1]   **Harlin Lee** [2]   **Mete Akgün** [1]

## Abstract

Unsupervised multi-source domain adaptation (UMDA) leverages labeled data from multiple source domains to generalize to an unlabeled target. While federated UMDA addresses privacy by avoiding raw data sharing, existing methods scale poorly as the number of sources increases, often suffering from high computational overhead or training instability. We propose GALA, a scalable and robust federated UMDA framework designed for high-diversity settings. GALA achieves scalability by coupling a novel inter-group discrepancy minimization objective that approximates pairwise alignment with linear complexity alongside a temperature-controlled, centroid-based weighting strategy for dynamic source prioritization. These components enable stable, parallelizable training across many heterogeneous sources, addressing a critical scalability bottleneck that remains largely unaddressed in current literature. To evaluate performance in high-diversity scenarios, we introduce Digit-18, a new benchmark comprising 18 datasets with varied synthetic and real-world domain shifts. Extensive experiments demonstrate that GALA achieves state-of-the-art results on standard benchmarks and significantly outperforms prior methods in large-scale settings where others either fail to converge or become computationally infeasible.

---

[*]Equal contribution  [1]Department of Computer Science, Medical Data Privacy and Privacy-Preserving Machine Learning (MDPPML) Group and Institute for Bioinformatics and Medical Informatics (IBMI), University of Tübingen, Tübingen, Germany  [2]School of Data Science and Society, University of North Carolina at Chapel Hill, NC, USA. Correspondence to: Cem Ata Baykara <cem.baykara@uni-tuebingen.de>, Larissa Reichart <larissa.reichart@uni-tuebingen.de>.

*Proceedings of the $43^{rd}$ International Conference on Machine Learning*, Seoul, South Korea. PMLR 306, 2026. Copyright 2026 by the author(s).

## 1. Introduction

Unsupervised multi-source domain adaptation (UMDA) (Zhang et al., 2015) aims to learn a model that generalizes to an unlabeled target domain by leveraging labeled data from multiple sources. Unlike single-source adaptation, multi-source setups better reflect real-world conditions where data is naturally distributed across diverse environments. However, the presence of domain shifts among sources, in addition to the shift to the target, makes multi-source adaptation substantially more challenging.

Prior work has shown that alignment of source and target structures can improve robustness to distributional shift (Chang et al., 2019; Zhao et al., 2020; Dai et al., 2020; Ganin & Lempitsky, 2015). Yet in privacy-sensitive domains such as healthcare, regulations like GDPR[1] and CCPA[2] restrict data sharing and require both computation and data remain local. This makes centralized training infeasible and motivates the use of distributed UMDA approaches, such as federated (Konečný et al., 2015; Smith et al., 2017) or decentralized (McMahan et al., 2017) learning.

Current distributed UMDA methods are limited in scalability when it comes to diverse multi-source settings. Most are designed for small-scale scenarios typically involving only 2 to 6 sources (Feng et al., 2021; Schrod et al., 2025; Liang et al., 2020; Peng et al., 2020). As the number of source domains increases, these methods either require prohibitive computational costs or suffer from degraded performance and unstable convergence.

In this paper, we propose **G**rouping-based **A**daptive **Lea**rning (**GALA**), a federated UMDA framework designed to scale with the number and heterogeneity of diverse source domains. GALA combines two key components: (1) an inter-group discrepancy minimization objective that aligns aggregated source predictions without computing all pairwise discrepancies (Section 3.3); and (2) a temperature-scaled centroid-based weighting scheme that dynamically estimates each source's alignment with the target domain (Section 3.4). By randomly partitioning sources into groups and minimizing disagreement between their weighted pre-

---

[1]General Data Protection Regulation, European Union
[2]California Consumer Privacy Act

dictions on the unlabeled target domain, GALA approximates the global alignment objective in a scalable and robust manner.

To evaluate multi-source settings more realistically than duplicating domains across clients, we introduce **Digit-18**, a benchmark of 18 digit datasets spanning synthetic and real-world shifts. Extensive experiments show that GALA not only matches or exceeds state-of-the-art performance on standard UMDA benchmarks, but also maintains stability and robustness as the number of diverse sources grows, where existing methods fail to converge or become computationally infeasible. Our contributions are as follows:

- To our knowledge, we are the first to explicitly address scalability in distributed UMDA with respect to the number of heterogeneous source domains, showing that existing approaches either fail to converge or become computationally infeasible as the number of sources grows.

- We propose Inter-Group Discrepancy (IGD), a group-level discrepancy objective that approximates the full pairwise disagreement at linear cost with low variance, enabling robust alignment when source count is large.

- We introduce GALA, a scalable federated UMDA algorithm that couples IGD with a temperature-scaled, centroid-based similarity weighting to prioritize target-relevant sources and mitigate negative transfer in high-diversity settings.

- We introduce Digit-18, a challenging large-scale UMDA benchmark designed to test scalability and source heterogeneity, and use it alongside standard datasets for comprehensive evaluation.

- Through extensive experiments and ablation studies, we demonstrate that GALA consistently improves accuracy, stability, and convergence compared to prior methods across both standard and high-diversity multi-source scenarios.

## 2. Related Work

**Unsupervised Multi-Source Domain Adaptation** Standard UMDA techniques aim to learn domain-invariant representations that generalize well to an unlabeled target domain by reducing discrepancies between the source and target distributions (Ben-David et al., 2010; Zhao et al., 2018). This is mainly achieved by two approaches: maximum mean discrepancy (MMD) (Tzeng et al., 2014; Peng et al., 2019a) and adversarial training (Saito et al., 2018; Peng et al., 2020; Liu et al., 2018).

Adversarial domain adaptation typically achieves alignment between source and target feature distributions by training a domain discriminator to make source and target feature distributions indistinguishable (Zhao et al., 2018). However, Zhao et al. (2019) and Liu et al. (2019) warn that aligning only input features can be insufficient or even harmful, and feature-level discrepancy minimization does not necessarily yield invariant representations and may increase the target error in some cases. In federated multi-source settings, this can be more problematic because discriminator training generally requires simultaneous access to samples from multiple domains.

**Federated Domain Adaptation** Federated learning is a distributed machine learning technique which allows collaborative training of a global model through aggregation of local model updates (Konečný et al., 2016). Data heterogeneity in federated learning arises from various forms of distribution mismatch across clients and remains a major challenge (Li et al., 2020; Karimireddy et al., 2020; Wang et al., 2023). Federated UMDA addresses one such setting by focusing on domain shifts across distributed clients. It was first proposed by Peng et al. (2020), which uses adversarial training to minimize $\mathcal{H}$-divergence without direct access to data. FACT (Schrod et al., 2025) is a recent approach that aligns source and target representations using inter-domain differences rather than adversarial training. It achieves state-of-the-art performance on digit datasets while being inherently scalable and efficient. However, while scalable, this approach introduces high variance and suffers from convergence issues when the number of sources grows, as each training step involves only a single pair of sources. Our empirical results show that FACT's performance becomes unstable in high-source scenarios and often fails to converge on challenging target domains.

**Decentralized Domain Adaptation** Decentralized UMDA methods resemble their federated counterparts, with the key difference that training is not coordinated through a central server (Wu & Gong, 2021). Liang et al. (2020) proposes a source-free strategy for single-source domain adaptation that can also be extended to multi-source settings. In SFDA (Wang et al., 2022), the Multi-Domain Model Generalization Balance (MDMGB) algorithm is introduced to adaptively weight multiple source models according to their similarity to the target domain. The target predictor is then trained separately using pseudo-labeling and information maximization. However, while it is efficient in terms of communication rounds, its performance on benchmark datasets is not state-of-the-art.

More recently, Yang et al. (2025) propose to fuse domain-invariant and domain-specific features, arguing that retaining domain-specific information is important for effective classification. Currently, KD3A by Feng et al. (2021) represents the state-of-the-art in decentralized UMDA, using

a consensus-driven alignment strategy that achieves strong accuracy and robustness against negative transfer across multiple benchmarks.

However, KD3A is inherently not scalable to large multi-source settings, as it requires per-domain optimization and divergence computation to be performed locally at the target. This makes it unsuitable for distributed scenarios where target access is limited or expensive. Our empirical analysis shows that even when source training is parallelized to mimic practical deployment, KD3A's computation time grows exponentially with the number of sources, becoming infeasible in large multi-source settings.

## 3. Methodology

**Preliminaries.** In a UMDA setting, we have $N$ distinct source domains $\{\mathbb{D}_S^n\}_{n=1}^N$ where each domain contains $K_n$ labeled samples as $\{\mathbb{D}_S^n\} := \{(x_i^n, y_i^n)\}_{i=1}^{K_n}$, and a target domain $\mathbb{D}_T$ with $K_T$ unlabeled samples $\mathbb{D}_T := \{x_i^T\}_{i=1}^{K_T}$. We consider a $C$-way classification task shared across all domains, and assume every domain contains samples from every class with roughly uniform numbers of samples per class. The main objective of UMDA is to learn a *feature extractor* $G : \mathcal{X} \to \mathbb{R}^d$, and a *predictor* $F : \mathbb{R}^d \to \Delta^C$, where $\Delta^C$ is the probability vector over $C$ classes. Together they define the model $h = F \circ G \in \mathcal{H}$ that minimizes the task error $\epsilon_{\mathbb{D}_T}(h) = \Pr_{(x,y)\sim\mathbb{D}_T}[h(x) \neq y]$. See Ben-David et al. (2010); Zhao et al. (2018) for formal definitions of $\mathcal{H}$-divergences $d_{\mathcal{H}}, d_{\mathcal{H}\Delta\mathcal{H}}$.

### 3.1. Federated UMDA Theoretical Motivation

For any set of weights $\{w_n\}$ that determines the contribution of each source domain to the final predictor, the following generalization bound holds for federated UMDA. It stems from a simple convex combination of the classical UMDA generalization bound (Zhao et al., 2018; Ben-David et al., 2010), so we do not make claims on novelty or tightness. We present this only to gain some insight into federated UMDA with large data heterogenity and motivate our approach.

**Corollary 3.1** (Generalization bound of federated UMDA). *Let* $w_1, \ldots, w_N \in \mathbb{R}_+$ *satisfy* $\sum_{n=1}^N w_n = 1$. $\mathbb{D}_{\tilde{S}}$ *is pooled sources, and* $p_n = K_n / \sum_{i=1}^N K_i$ *is relative sample size of* $\mathbb{D}_S^n$ *compared to* $\mathbb{D}_{\tilde{S}}$. *Then for any* $h \in \mathcal{H}$, *w.h.p.*

$$\epsilon_{\mathbb{D}_T}(h) \leq \hat{\epsilon}_{\mathbb{D}_{\tilde{S}}}(h) + \sum_{n=1}^N (w_n - p_n)\hat{\epsilon}_{\mathbb{D}_S^n}(h) \qquad (1)$$

$$+ \frac{1}{2}\boxed{\sum_{n=1}^N w_n\, d_{\mathcal{H}\Delta\mathcal{H}}(\mathbb{D}_S^n, \mathbb{D}_T)} + \sum_{n=1}^N w_n\lambda_n + \tilde{O}\left(\frac{1}{K_T}\right),$$

*where* $\lambda_n := \min_h \epsilon_{\mathbb{D}_S^n}(h) + \epsilon_{\mathbb{D}_T}(h)$.

This highlights the effect of federated learning. The first term is the empirical loss one would obtain if all source data were pooled, and the second term isolates the deviation introduced by source heterogeneity and the use of $w_n$ instead of sample-proportion weights, becoming positive or negative depending on whether clients contribute unevenly relative to their dataset sizes.

This also makes explicit that the optimal joint error term depends on $w_n$, so the bound cannot be directly minimized in practice to identify the "best" $w_n$ or single source. However, the boxed divergence term does suggest that aligning source feature distributions with the target may lead to reduced target error and can be attempted without labels. GALA fits in this line of work.

In the following sections, we design and compute a scalable proxy loss $\mathcal{L}_{\mathrm{IGD}}$ for the boxed divergence term.

### 3.2. Prior Work: Minimizing Divergence for UMDA

Schrod et al. (2025) recently proposed to measure predictor disagreement on unlabeled target samples to quantify inter-source differences and minimize the average pairwise disagreement between source predictors. This idea is captured by the full pairwise discrepancy:

$$\mathcal{L}_{\mathrm{full}} = \sum_{i<j} \mathbb{E}_{x\sim\mathbb{D}_T}\left[\|F_i(G(x)) - F_j(G(x))\|_1\right], \qquad (2)$$

where the sum $\sum_{i<j}$ ranges over all unordered pairs of distinct source domains $i, j \in \{1, \ldots, N\}$. An intuitive explanation of the connection between Corollary 3.1 and the proxy loss is deferred to Appendix A.

Because this formulation scales quadratically with the number of sources, requiring $\binom{N}{2}$ pairwise comparisons, it becomes impractical in large-scale settings. To address this, Schrod et al. (2025) further introduces the concept of Inter-Domain Distance (IDD) minimization, where, at each round, two source domains are randomly selected, and the disagreement between their predictors on target data is minimized. This is formally defined as:

$$\mathcal{L}_{\mathrm{IDD}}^{(i,j)} = \mathbb{E}_{x\sim\mathbb{D}_T}\left[\|F_i(G(x)) - F_j(G(x))\|_1\right], \qquad (3)$$

which encourages domain-invariant representations by aligning the outputs of individual source predictors on the target distribution. Although efficient and unbiased, this approach introduces high variance, as each update reflects only the behavior of a single random pair of sources. Our empirical findings show that this approach becomes unstable in diverse multi-source settings and frequently fails to converge on challenging target domains.

### 3.3. Achieving Scalability through Inter-Group Discrepancy

To mitigate these issues, we introduce Inter-Group Discrepancy (IGD), a group-level discrepancy objective that serves as an efficient, low-variance approximation to $\mathcal{L}_{\text{full}}$ without requiring full pairwise computation or reliance on a single random pairing. At each training round, IGD randomly partitions the $N$ source predictors $\{F_n\}_{n=1}^N$ into two disjoint groups $\mathcal{G}_1$ and $\mathcal{G}_2$, forms weighted average predictors for each group, and minimizes their $\ell_1$ disagreement on unlabeled target data:

$$\mathcal{L}_{\text{IGD}} = \mathbb{E}_{x \sim \mathbb{D}_T}\left[\|F_{\mathcal{G}_1}(G(x)) - F_{\mathcal{G}_2}(G(x))\|_1\right], \quad (4)$$

with group predictors:

$$F_{\mathcal{G}_1} = \sum_{i \in \mathcal{G}_1} \tilde{w}_i F_i, \quad F_{\mathcal{G}_2} = \sum_{j \in \mathcal{G}_2} \tilde{w}_j F_j, \quad (5)$$

$$\text{where} \sum_{i \in \mathcal{G}_k} \tilde{w}_i = 1, \quad \tilde{w}_i \geq 0 \quad \text{for } k = 1, 2.$$

**Intuition.** IDD and IGD both aim to estimate $\mathcal{L}_{\text{full}}$, which is expensive to compute. IDD is an efficient and unbiased but high-variance estimator of $\mathcal{L}_{\text{full}}$. As for IGD, a quick manipulation of Eq. 4 and 5 shows that IGD is equivalent to

$$\mathbb{E}_{x \sim \mathbb{D}_T}\left[\left\|\sum_{i \in \mathcal{G}_1}\sum_{j \in \mathcal{G}_2} \tilde{w}_i \tilde{w}_j \left(F_i(G(x)) - F_j(G(x))\right)\right\|_1\right].$$

Intuitively, by averaging many pairwise differences before taking the $\ell_1$ norm, IGD yields lower variance than the single-pair discrepancy estimator such as IDD in Eq. 3. Explicitly, we show in Appendix B that IGD is a biased but low-variance estimator of the full pairwise objective, both theoretically under uniform weights and empirically under GALA weights. In practice, our extensive experiments show that the effect of this bias on GALA is small, and the resulting variance reduction leads to more stable training and improved performance in highly diverse multi-source settings.

### 3.4. Enhancing IGD through Source Relevance

The practical effectiveness of IGD depends on how much influence each source exerts in the group aggregates, i.e. the weights. If group predictors place large importance on irrelevant or noisy sources, the aggregated outputs can be biased and drive negative transfer rather than alignment. Therefore, in addition to minimizing group disagreement,

we require a lightweight and private mechanism that (i) estimates a global relevance score $w_n$ for each source (so that well-aligned sources contribute more to the final predictor), and (ii) produces group-normalized weights $\tilde{w}_n$ so that IGD focuses its alignment on relevant sources within each partition.

**Sources Weighting via Centroid Similarity.** To estimate similarity between each source domain and the target without accessing labels or sharing data, we adopt a centroid-based proxy inspired by the MDMGB algorithm of Wang et al. (2022). MDMGB computes similarity using class-wise centroids in feature space. Specifically, each domain computes a soft centroid for class $c$ as:

$$r_n^c = \frac{\sum_{x \in \mathbb{D}_S^n} \delta_c(x) \cdot G(x)}{\sum_{x \in \mathbb{D}_S^n} \delta_c(x)}, \quad r_T^c = \frac{\sum_{x \in \mathbb{D}_T} \delta_c(x) \cdot G(x)}{\sum_{x \in \mathbb{D}_T} \delta_c(x)},$$

where $\delta_c(x)$ is the softmax probability for class $c$ from predictor $F$. A cosine similarity score between source and target centroids is then computed by:

$$S(r_T, r_n) = \sum_{c=1}^C \frac{\langle r_T^c, r_n^c \rangle}{\|r_T^c\|\|r_n^c\|} + 1. \quad (6)$$

**Limitations of MDMGB in Diverse Source Settings.** While effective in low-diversity settings, the original MD-MGB approach underperforms when source domains vary widely in quality or distribution. In such cases, the computed similarities fail to sharply penalize misaligned domains, resulting in negative transfer and poor target alignment. Our ablation analysis show that using unmodified MDMGB in diverse multi-source scenarios performs comparably to using uniform weights.

To address this limitation, we propose a modified version of MDMGB, which we refer to as MDMGB+. This variant introduces a softmax-based selection mechanism with a tunable temperature parameter $\tau > 0$ that sharpens the similarity contrast among source domains. We define the global relevance score for each source as:

$$w_n = \frac{\exp(\tau \cdot S(r_T, r_n))}{\sum_{j=1}^N \exp(\tau \cdot S(r_T, r_j))}. \quad (7)$$

Finally, within each partition $\mathcal{G}_k$, these global scores are re-normalized to form group-specific weights:

$$\tilde{w}_n = \frac{w_n}{\sum_{j \in \mathcal{G}_k} w_j}, \quad n \in \mathcal{G}_k, \ k \in \{1, 2\}. \quad (8)$$

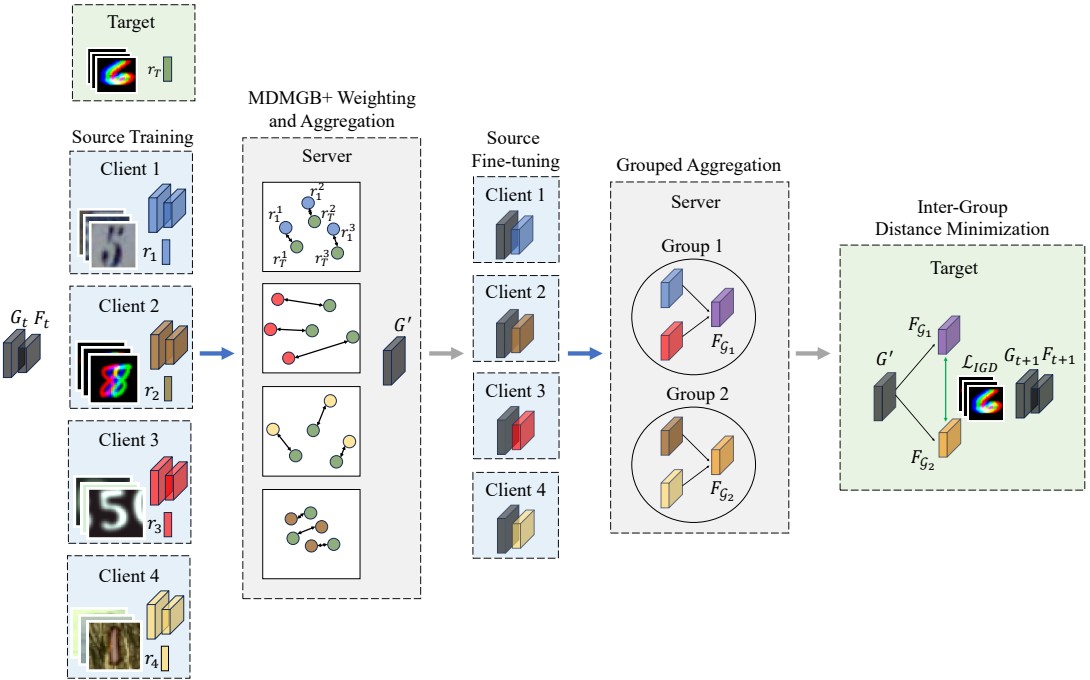

*Figure 1.* Visual overview of the GALA framework (Algorithm 1) for federated unsupervised multi-domain adaptation.

The temperature parameter $\tau$ controls the selectivity of the weighting: higher values amplify small similarity differences, assigning more importance to sources better aligned with the target. MDMGB+ enables IGD to remain effective in diverse multi-source settings, where relevant domains might otherwise be dominated by dissimilar ones.

### 3.5. The GALA Algorithm

GALA combines the IGD objective and MDMGB+ relevance weighting into a federated process that iteratively aligns source predictors with an unlabeled target. A high-level overview is shown in Figure 1. At round $t$, the current global model is $h_t = F_t \circ G_t$. Source and target clients compute class-wise soft centroids in feature space ($r_n^c$ and $r_T^c$) from local data, and share only these centroid summaries with the server. Clients also perform local training updates of the global model and transmit the updated parameters to the server.

Using the received centroids, the server computes a global relevance profile $\{w_n\}$ via the MDMGB+ softmax (Eq. 7) which differentiates between well-aligned and misaligned sources and yields group-normalized weights $\{\tilde{w}_n\}$. The relevance scores are used to aggregate the client's feature

extractors to a consensus feature extractor $G'$. Clients then locally fine-tune their predictors to maintain compatibility with $G'$.

To align the consensus extractor with the unlabeled target, the fine-tuned predictors are randomly partitioned into two groups to form group predictors $F_{\mathcal{G}_1}, F_{\mathcal{G}_2}$ (Eq. 5). The extractor is then updated on the target data by minimizing the IGD loss (Eq. 4) to obtain $G''$. Finally, the group predictors are merged into a single relevance-weighted global predictor $F_{t+1}$, and the next global model is defined as $h_{t+1} = F_{t+1} \circ G_{t+1}$ with $G_{t+1} = G''$. This cycle repeats until convergence. The step-by-step algorithm and implementation details are provided in Appendix C.

## 4. Experiments

We evaluate GALA on standard UMDA benchmarks, study its scalability as the number of distinct source domains increases, and analyze its training efficiency in federated settings. Our evaluation considers four datasets:

**DomainNet.** (Peng et al., 2019b) A widely used domain adaptation benchmark with 345 classes across six domains,

*Table 1.* UMDA accuracy (%) on the DomainNet dataset. $^{*}$: Re-implementation results reported in Feng et al. (2021). $^{\dagger}$: Without data augmentation. We report mean and standard deviation over five runs. Best results among **distributed** methods are shown in bold, and second-best results are underlined.

| Setting | Method | Clipart | Infograph | Painting | Quickdraw | Real | Sketch | Avg |
|---------|--------|---------|-----------|----------|-----------|------|--------|-----|
| W/o DA | Oracle | $69.3_{\pm0.37}$ | $34.5_{\pm0.42}$ | $66.3_{\pm0.67}$ | $66.8_{\pm0.51}$ | $80.1_{\pm0.59}$ | $60.7_{\pm0.48}$ | 63.0 |
| | Source-only | $52.1_{\pm0.51}$ | $23.1_{\pm0.28}$ | $47.7_{\pm0.96}$ | $13.3_{\pm0.72}$ | $60.7_{\pm0.32}$ | $46.5_{\pm0.56}$ | 40.6 |
| Centralized | MDAN | $60.3_{\pm0.41}$ | $25.0_{\pm0.43}$ | $50.3_{\pm0.36}$ | $8.2_{\pm1.92}$ | $61.5_{\pm0.46}$ | $51.3_{\pm0.58}$ | 42.8 |
| | $M^3$SDA | $58.6_{\pm0.53}$ | $26.0_{\pm0.89}$ | $52.3_{\pm0.55}$ | $6.3_{\pm0.58}$ | $62.7_{\pm0.51}$ | $49.5_{\pm0.76}$ | 42.6 |
| | CMSS | $64.2_{\pm0.18}$ | $28.0_{\pm0.2}$ | $53.6_{\pm0.39}$ | $16.0_{\pm0.12}$ | $63.4_{\pm0.21}$ | $53.8_{\pm0.35}$ | 46.5 |
| | DSBN* | 60.3 | 22.6 | 52.3 | 9.1 | 62.7 | 47.6 | 42.4 |
| | DAEL | $70.8_{\pm0.14}$ | $26.5_{\pm0.13}$ | $57.4_{\pm0.28}$ | $12.2_{\pm0.7}$ | $65.0_{\pm0.23}$ | $60.6_{\pm0.25}$ | 48.7 |
| Distributed | FADA* | 59.1 | 21.7 | 47.9 | 8.8 | 60.8 | 50.4 | 41.5 |
| | SHOT* | 61.7 | **22.2** | 52.6 | 12.2 | 67.7 | 48.6 | 44.2 |
| | KD3A$^{\dagger}$ | $\underline{69.7}_{\pm0.67}$ | $21.2_{\pm0.35}$ | **$58.8_{\pm0.66}$** | $\underline{15.1}_{\pm0.21}$ | **$70.4_{\pm0.54}$** | **$57.9_{\pm0.41}$** | $\underline{48.8}$ |
| | GALA | **$71.1_{\pm0.18}$** | $\underline{21.9}_{\pm0.20}$ | $\underline{58.1}_{\pm0.14}$ | **$19.2_{\pm0.20}$** | **$70.4_{\pm0.12}$** | $\underline{57.7}_{\pm0.08}$ | **49.7** |

designed to test highly heterogeneous domain shifts.

**Office-Caltech10.** (Saenko et al., 2010; Griffin et al., 2022) A small-scale object recognition benchmark with four domains and 10 shared categories.

**Digit-Five.** (Peng et al., 2019a) A standard benchmark with five digit domains and moderate source-target shifts.

**Digit-18 (ours).** A new large-scale benchmark comprising 18 diverse digit domains. These domains were created by systematically applying techniques such as background augmentation, scaling, and color channel stacking to existing digit datasets, resulting in substantial distributional shifts. Full details on dataset generation and inter-domain similarity analysis are provided in Appendix E.

**Baselines.** We compare GALA against both centralized and federated UMDA baselines. Central baselines include MDAN (Zhao et al., 2018), M$^3$SDA (Peng et al., 2019a), CMSS (Yang et al., 2020), DAEL (Zhou et al., 2021), DSBN (Chang et al., 2019), and DANE (Yang et al., 2024). Distributed baselines include SHOT (Liang et al., 2020), FADA (Peng et al., 2020), SFDA (Wang et al., 2022), FACT (Schrod et al., 2025), and KD3A (Feng et al., 2021). We also report a source-only baseline, which is the model trained only on labeled source domains and evaluated on the target without any adaptation, and an Oracle baseline, which is the model trained and validated with access to labeled target domain data.

**Implementation Details.** For the digit datasets (Digit-Five and Digit-18), we use a 2-layer CNN with two convolutional blocks followed by three fully connected layers, with dropout and batch normalization. For DomainNet

and Office-Caltech10, we adopt a ResNet101 pretrained on ImageNet (see Appendix D for the full architecture). All models are trained with SGD (momentum 0.9, weight decay $5 \times 10^{-4}$). The temperature parameter $\tau$ is set to 0.03 for Clipart and Sketch targets and 0.05 for all other Domain-Net targets, 1.0 for Office-Caltech10 and Digit-18, and 0.2 for Digit-Five. For DomainNet, Digit-Five, and Digit-18, we use a custom learning-rate scheduler with decay factor $\gamma = 0.75$; for Office-Caltech10, we use exponential decay with $\gamma = 0.9$. Communication occurs once per epoch ($r = 1$), with one epoch for each stage: source training, source fine-tuning, and inter-group discrepancy minimization. We report mean $\pm$ std accuracy over five runs using an AMD EPYC 7713 CPU and an NVIDIA A100 GPU (40GB). The code for reproducing the experiments, as well as the Digit-18 benchmark, is available on GitHub[3].

### 4.1. Performance on Standard Benchmarks

**DomainNet.** As shown in Table 1, GALA achieves strong performance on DomainNet, even though this benchmark contains only 6 domains and is not our primary focus compared with the higher-source settings considered in the main experiments. GALA achieves the best average accuracy of 49.7%, outperforming KD3A by 0.9% and slightly improving over the best centralized baseline, DAEL (48.7%). In particular, GALA obtains the best results on Clipart (71.1%), Infograph (21.9%), and Quickdraw (19.2%), while matching the best performance on Real (70.4%) and remaining competitive on Painting and Sketch. These results indicate that, although DomainNet is a relatively smaller multi-source setting in terms of number of sources, GALA generalizes well beyond the digit benchmarks and continues to

---

[3]github.com/mdppml/GALA.git

*Table 2.* UMDA accuracy (%) on various target domains on the Digit-18 dataset. We report mean and standard deviation over five runs. Best results among distributed methods are shown in bold.

| Method | *mnist* | *mnistm* | *svhn* | *syn* | *usps* |
|---|---|---|---|---|---|
| Oracle | $99.0_{\pm0.03}$ | $95.6_{\pm0.26}$ | $88.4_{\pm0.15}$ | $97.0_{\pm0.12}$ | $98.9_{\pm0.12}$ |
| FACT | $98.6_{\pm0.20}$ | $87.6_{\pm1.58}$ | $92.5_{\pm0.41}$ | $97.4_{\pm0.46}$ | $98.3_{\pm0.26}$ |
| GALA | $\mathbf{99.3}_{\pm0.07}$ | $\mathbf{95.2}_{\pm0.10}$ | $\mathbf{95.4}_{\pm0.17}$ | $\mathbf{98.4}_{\pm0.05}$ | $\mathbf{99.0}_{\pm0.10}$ |
| | ↑ **0.7** | ↑ **7.6** | ↑ **2.9** | ↑ **1.0** | ↑ **0.7** |
| Method | *synm* | *svhn-xs* | *svhnstack* | *usps-m* | *ardis-xs* | Avg |
| Oracle | $83.8_{\pm0.32}$ | $84.7_{\pm0.37}$ | $86.5_{\pm0.04}$ | $92.0_{\pm0.51}$ | $97.5_{\pm0.32}$ | 92.3 |
| FACT | $79.4_{\pm1.51}$ | $79.1_{\pm4.75}$ | $91.9_{\pm1.63}$ | $87.6_{\pm1.06}$ | $93.9_{\pm1.21}$ | 90.6 |
| GALA | $\mathbf{85.8}_{\pm0.23}$ | $\mathbf{88.6}_{\pm0.29}$ | $\mathbf{94.8}_{\pm0.10}$ | $\mathbf{92.9}_{\pm0.34}$ | $\mathbf{97.1}_{\pm0.21}$ | **94.6** |
| | ↑ **6.4** | ↑ **9.5** | ↑ **2.9** | ↑ **5.3** | ↑ **3.2** | ↑ **4.0** |

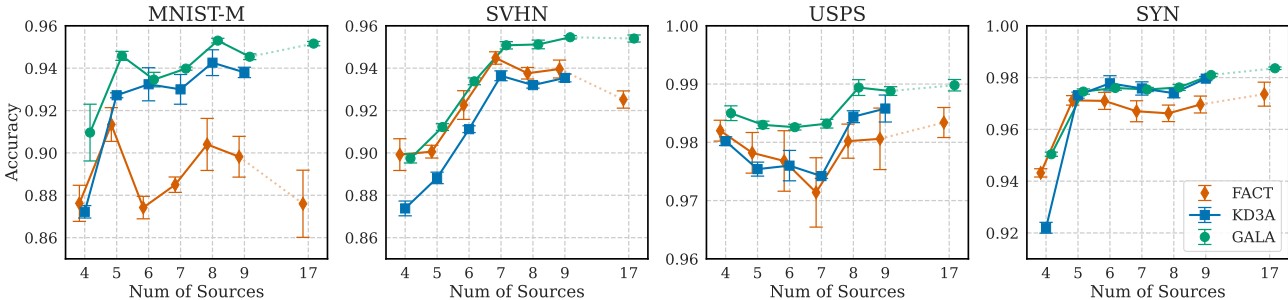

*Figure 2.* UMDA accuracy on Digit-Five targets for increasing number of source domains. KD3A is excluded beyond 9 sources due to exponential runtime.

provide strong performance in complex, higher-dimensional settings.

**Digit-Five and Office-Caltech10.** The results for Digit-Five and Office-Caltech10 are provided in Appendix F. On Digit-Five, GALA achieves the highest average accuracy and performs best on most target domains, while on the remaining targets it is only marginally below the best method, with differences of around 0.2%. On Office-Caltech10, GALA also achieves competitive performance and ranks among the top methods on this small benchmark. In particular, it performs very strongly on DSLR and Webcam, including 100% on DSLR. Overall, these results demonstrate that GALA matches or surpasses state-of-the-art methods on standard benchmarks, serving as a strong baseline for the more challenging settings considered next.

### 4.2. Scalability under Growing Source Diversity

Next, we evaluate performance as the number of source domains increases. Starting from the 4-source Digit-Five setup, we progressively add Digit-18 domains by increasing task difficulty (lowest self-performance first; see Appendix E.1.3). Figure 2 shows results across Digit-Five targets.

While all methods initially benefit from additional source do-

mains, performance diverges as dissimilar or noisy sources are added. FACT becomes unstable beyond 9 sources, showing higher variance due to its reliance on randomly sampled pairs. KD3A remains robust but suffers exponential runtime growth, making it impractical for high-source settings. In contrast, GALA maintains stable accuracy with dynamic weighting that suppresses negative transfer. Across all targets, GALA improves with more sources and consistently outperforms KD3A and FACT. Due to KD3A's runtime, the full 18-domain evaluation is restricted to FACT and GALA.

**Full Digit-18 Results.** Table 2 reports accuracy across 10 target domains in the full 18-source setting, highlighting the challenges of this high-diversity scenario. GALA achieves a 4.0% average gain over FACT and outperforms all baselines on every target. Improvements are particularly pronounced on the most challenging domains, such as SVHNXS (+9.5%) and SYNM (+6.4%), where FACT struggles but GALA still maintains strong performance. Notably, GALA even surpasses the oracle on some targets, indicating that it effectively leverages information across domains to improve generalization. These results show that in multi-source settings with more than ten sources, GALA overcomes prior limitations and establishes a new state-of-the-art across all targets.

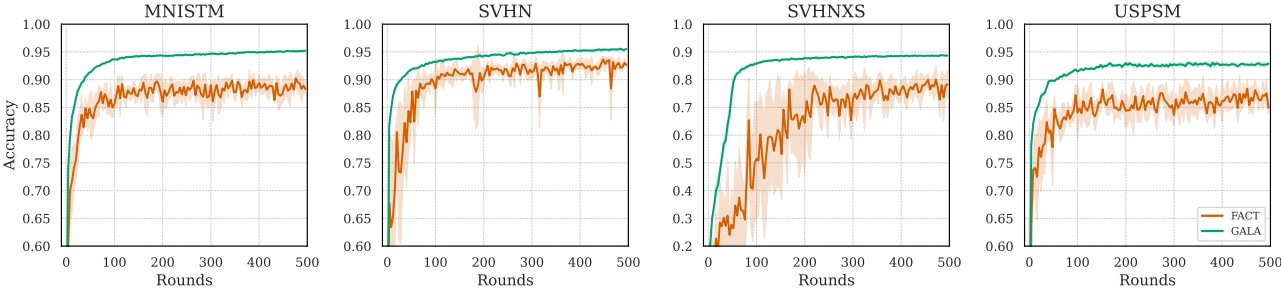

*Figure 3.* UMDA accuracy over training rounds on the MNISTM, SVHN, SVHNXS, and USPSM targets on the Digit-18 dataset.

**Training Dynamics.** Figure 3 shows test accuracy over training rounds for four challenging target domains. GALA converges quickly, surpassing FACT's final accuracy within the first 100 rounds, and continues to improve slightly thereafter. Accuracy remains stable both between rounds and across 10 runs. In contrast, FACT exhibits large fluctuations and struggles on difficult targets such as MNISTM, SVHN, and SVHNXS. Additional results on easier targets are provided in Appendix I, where the same trend appears but is less pronounced.

### 4.3. Runtime Comparison

To evaluate computational efficiency, we compare per-round runtimes under an idealized federated setting where client-side operations are parallelized to simulate practical execution without bandwidth constraints (Table 3).

*Table 3.* Per-round training time (in seconds) for varying numbers of source domains.

| # Sources | 3 | 5 | 7 | 9 |
|---|---|---|---|---|
| KD3A | $50.73_{\pm 0.3}$ | $216.03_{\pm 2.7}$ | $1029.84_{\pm 10.6}$ | $5600.48_{\pm 32.0}$ |
| FACT | $3.65_{\pm 0.3}$ | $3.22_{\pm 0.4}$ | $3.39_{\pm 0.5}$ | $4.05_{\pm 0.5}$ |
| GALA | $17.27_{\pm 0.1}$ | $17.79_{\pm 0.7}$ | $22.21_{\pm 0.1}$ | $22.37_{\pm 0.1}$ |

For KD3A, while initial source training is parallelized, the subsequent consensus and knowledge-voting stages are executed sequentially on the target (Feng et al., 2021), forcing all sources to remain idle during this phase. In contrast, GALA and FACT parallelize all local training and fine-tuning steps. We report the maximum per-source runtime per round as a practical upper bound, averaged over five runs to mitigate hardware fluctuations. As shown in the results, KD3A scales poorly because its consensus step requires computing source permutations, leading to exponential runtime growth as the number of domains increases. GALA and FACT avoid this bottleneck, maintaining high efficiency. While GALA incurs a slightly higher per-round cost than FACT due to the inter-group alignment, it avoids the computational infeasibility seen in KD3A in diverse, large-scale multi-domain scenarios.

### 4.4. Parameter Analysis: $\tau$ in MDMGB+

We evaluate the sensitivity of MDMGB+ to the temperature parameter $\tau$, which controls the sharpness of source relevance weighting. Figure 4 reports the final target accuracy (mean over five runs) for different values of $\tau$, with (a) corresponding to Digit-Five and (b) to Digit-18 sources.

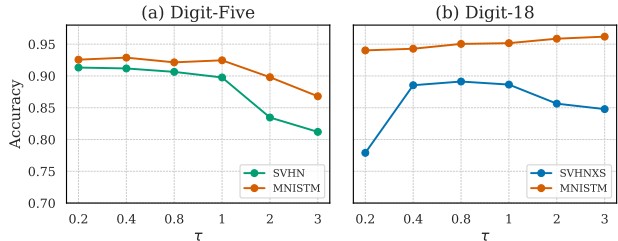

*Figure 4.* Effect of $\tau$ in GALA on UMDA performance for SVHN and MNIST-M as targets in the Digit-Five dataset, and SVHNXS and MNIST-M as targets in the Digit-18 dataset.

On Digit-18, very low temperatures (e.g., $\tau = 0.2$) produce overly uniform weights, limiting the model's ability to focus on well-aligned sources, as reflected by reduced accuracy. In contrast, excessively high values (e.g., $\tau = 3$) result in lower final performance despite initially faster convergence. Intermediate values ($\tau \in [0.8, 1.0]$) offer the best trade-off, yielding stable and accurate performance.

On Digit-Five, which exhibits lower source diversity, lower to moderate temperatures ($\tau \in [0.4, 1.0]$) achieve the highest accuracy, while larger values tend to degrade performance. This behavior is consistent with the fact that many source domains are relevant in this setting, and overly sharp weighting can unnecessarily overemphasize a small subset.

Overall, these results suggest that moderate $\tau$ values provide an effective default across benchmarks, while additional knowledge about source diversity or access to validation data can be leveraged to further fine-tune $\tau$ for additional performance gains.

## 4.5. Ablation Study

To isolate the contribution of our method, we evaluate the following configurations of GALA under the full 17-source Digit-18 setup: (1) IGD with uniform source contributions (no weighting), (2) IGD combined with MDMGB (Wang et al., 2022), (3) IGD combined with our proposed MD-MGB+ with $\tau = 1.0$, (4) GALA without target training (no IGD) combined with MDMGB+ ($\tau = 1.0$), and (5) GALA without target training and without weighting.

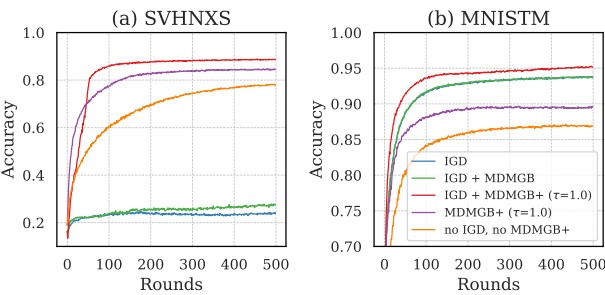

*Figure 5.* Effect of IGD and weighting strategies (no weighting, MDMGB, and MDMGB+) on UMDA performance for SVHNXS and MNIST-M as targets in the Digit-18 dataset.

Figure 5 reports results for two challenging targets, MNISTM and SVHNXS. On MNISTM, each component of GALA contributes positively, with IGD being more important than weighting. On SVHNXS, by contrast, domain selection with MDMGB+ is crucial. IGD alone fails to converge, MDMGB offers almost no improvement, and variants without MDMGB+ remain weaker. Overall, the full combination of IGD and MDMGB+ yields the most stable convergence and strongest performance by effectively capturing domain relevance under high source diversity.

## 4.6. Limitations

While GALA scales effectively to many source domains, it incurs higher computational and communication costs per round since all sources participate in training and fine-tuning. This full participation, though parallelizable, contrasts with more selective methods like FACT. Future work could reduce source participation or communication frequency to improve efficiency. Although we introduce Digit-18 to address the scarcity of public benchmarks with many diverse sources, our experiments focus on digit datasets. Extending evaluation to broader domains remains an important direction.

## 5. Conclusion

We introduced GALA, a federated framework for unsupervised multi-source domain adaptation that addresses the scalability challenges of diverse multi-source settings.

By combining temperature-scaled centroid-based weighting with inter-group discrepancy minimization, GALA enables robust, efficient alignment of diverse source domains to an unlabeled target. Our method achieves state-of-the-art performance across standard UMDA benchmarks and demonstrates strong stability and accuracy in large-scale settings where existing approaches degrade or fail to converge. Through our new Digit-18 benchmark, we further validate GALA's effectiveness under realistic, high-diversity conditions.

## Acknowledgements

This research was supported by the German Federal Ministry of Education and Research (BMBF) under project numbers 01ZZ2010 (MDPPML) and 01ZZ2316D (PrivateAIM). The authors thank the University of North Carolina at Chapel Hill, the Research Computing group, and the Training Center for Machine Learning (TCML) at the University of Tübingen for providing computational resources and support that contributed to this work.

## Impact Statement

This paper presents work whose goal is to advance the field of Machine Learning. There are many potential societal consequences of our work, none which we feel must be specifically highlighted here.

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

# Appendix

## A. Connection between the Divergence in Corollary 3.1 and the Loss Functions

Here we make explicit how the divergence term in Corollary 3.1 relates to $\mathcal{L}_{\text{full}}$, $\mathcal{L}_{\text{IDD}}^{(i,j)}$ and $\mathcal{L}_{\text{IGD}}$. Recall the definition of $d_{\mathcal{H}\Delta\mathcal{H}}(\mathbb{D}_S, \mathbb{D}_T)$ for binary classification:

$$2 \sup_{h,h' \in \mathcal{H}} \left| \Pr_{x \sim \mathbb{D}_S}[h(x) \neq h'(x)] - \Pr_{x \sim \mathbb{D}_T}[h(x) \neq h'(x)] \right|. \tag{9}$$

Intuitively, over pairs $h, h' \in \mathcal{H}$ we (i) find the region where they disagree, (ii) measure the mass that $\mathbb{D}_S$ and $\mathbb{D}_T$ place on that region, and (iii) take the largest (absolute) difference.

To obtain a practical, empirical estimator, we restrict $\mathcal{H}$ to the classifiers we actually have at a given round, i.e $F_1 \circ G, \ldots, F_N \circ G$. For multiclass tasks, we relax the binary disagreement to the $\ell_1$ difference of predicted probability vectors. Then we can see that Eq. 2 and 3 capture the same target-side disagreement notion as $\Pr_{x \sim \mathbb{D}_T}[h(x) \neq h'(x)]$.

Prior DA work such as (Saito et al., 2018) often assumes that source-side disagreement is negligible because each $F_n \circ G$ is well-trained on its own source, i.e. $\Pr_{x \sim \mathbb{D}_S}[h(x) \neq h'(x)] \approx 0$. We do not require this strong assumption because of source heterogeneity. Instead, we make the milder assumption that a well-trained $G$ allows $\Pr_{x \sim \mathbb{D}_S}[h(x) \neq h'(x)] \ll \Pr_{x \sim \mathbb{D}_T}[h(x) \neq h'(x)]$, which we find holds empirically. Under this assumption, the divergence $d_{\mathcal{H}\Delta\mathcal{H}}$ is dominated by the target-side disagreement, so minimizing the target-side disagreement terms provides an effective proxy for the divergence term in Corollary 3.1.

## B. IGD as a Biased but Low-variance Estimator of $\mathcal{L}_{\text{full}}$

IGD and IDD both aim to estimate $\mathcal{L}_{\text{full}}$, which is expensive to compute. IDD by Schrod et al. (2025) is an unbiased but high-variance estimator of $\mathcal{L}_{\text{full}}$. We now compare the bias and variance of IGD to those of IDD to motivate IGD.

Using the group predictors defined in Eq. 5, the IGD objective can be written as

$$\begin{aligned}
\mathcal{L}_{\text{IGD}} &= \mathbb{E}_{x \sim \mathbb{D}_T}\left[ \|F_{\mathcal{G}_1}(G(x)) - F_{\mathcal{G}_2}(G(x))\|_1 \right] \\
&= \mathbb{E}_{x \sim \mathbb{D}_T}\left[ \left\| \sum_{i \in \mathcal{G}_1} \tilde{w}_i F_i(G(x)) - \sum_{j \in \mathcal{G}_2} \tilde{w}_j F_j(G(x)) \right\|_1 \right] \\
&= \mathbb{E}_{x \sim \mathbb{D}_T}\left[ \left\| \sum_{i \in \mathcal{G}_1} \sum_{j \in \mathcal{G}_2} \tilde{w}_i \tilde{w}_j (F_i(G(x)) - F_j(G(x))) \right\|_1 \right].
\end{aligned}$$

Furthermore, Jensen's inequality yields

$$\mathcal{L}_{\text{IGD}} \leq \sum_{i \in \mathcal{G}_1} \sum_{j \in \mathcal{G}_2} \tilde{w}_i \tilde{w}_j E_{x \sim \mathbb{D}_T}\left[ \|F_i(G(x)) - F_j(G(x))\|_1 \right] = \sum_{i \in \mathcal{G}_1} \sum_{j \in \mathcal{G}_2} \tilde{w}_i \tilde{w}_j \, \mathcal{L}_{\text{IDD}}^{(i,j)} \leq \max_{i,j} \mathcal{L}_{\text{IDD}}^{(i,j)}. \tag{10}$$

Intuitively, because IGD averages many pairwise terms before taking the norm, its variance across random splits is smaller than that of an estimator that uses only a single pair $(i,j)$ per update, provided the weights are non-adversarial. While the observations above hold in general, we make the bias-variance tradeoff explicit under a simpler setting.

### B.1. Theoretical Bias and Variance of IGD under Uniform Weights

For exposition, assume we have even $N = 2M$ source predictors $F_1, \ldots, F_{2M}$ and form a random balanced partition into two groups $\mathcal{G}_1, \mathcal{G}_2$ of size $M$ each. Further assume uniform group-internal weights $\tilde{w}_i = 1/M$ for $i \in \mathcal{G}_k$. Finally, let's assume binary classification.

**Bias.** Define the normalized full pairwise discrepancy:

$$\text{(normalized) } \mathcal{L}_{\text{full}} = \frac{1}{M(2M-1)} \sum_{i<j} \mathbb{E}_{x \sim \mathbb{D}_T} \left[ \| F_i(G(x)) - F_j(G(x)) \|_1 \right].$$

Taking expectation (over random splits) on both sides of the inequality of Eq. 10 gives:

$$\mathbb{E}_{\text{split}}[\mathcal{L}_{\text{IGD}}] \leq \mathbb{E}_{\text{split}} \left[ \frac{1}{M^2} \sum_{i \in \mathcal{G}_1} \sum_{j \in \mathcal{G}_2} \mathcal{L}_{\text{IDD}}^{(i,j)} \right].$$

For any unordered pair $(i, j)$, the probability they fall into different groups is $P_{ij} = \frac{M}{2(2M-1)}$. Therefore

$$\mathbb{E}_{\text{split}}[\mathcal{L}_{\text{IGD}}] \leq \mathbb{E}_{\text{split}} \left[ \frac{1}{M^2} \sum_{i \in \mathcal{G}_1} \sum_{j \in \mathcal{G}_2} \mathcal{L}_{\text{IDD}}^{(i,j)} \right] = \frac{1}{M^2} \sum_{i \neq j} P_{ij} \cdot \mathcal{L}_{\text{IDD}}^{(i,j)}$$

$$= \frac{1}{M^2} \cdot 2 \sum_{i<j} \frac{N}{2(2M-1)} \mathcal{L}_{\text{IDD}}^{(i,j)} = \text{(normalized) } \mathcal{L}_{\text{full}}.$$

**Variance.** We will decompose the $\ell_1$ norm as sum of coordinates. We begin by noticing that there will be one coordinate of $F_i(G(x)) - F_j(G(x))$ that is in $[0, 1]$. We denote that variable $h_a(x) - h_b(x)$, where $a, b \in \{i, j\}$. Naturally the remaining coordinate would be $(1 - h_a(x)) - (1 - h_b(x)) = h_b(x) - h_a(x) \in [-1, 0]$, because $F$ maps to a probability vector. Further denote $\bar{h}_\mathcal{G}(x) = \sum_{i \in \mathcal{G}} h_i(x)$, and $\bar{h}(x)$ the average of all $h_i(x)$.

We aim to compare the following quantities:

$$\text{Variance of IDD: } \mathbb{V}_{a,b} \left[ \mathbb{E}_x \left[ 2(h_a(x) - h_b(x)) \right] \right] = 8 \mathbb{V}_k \left[ \mathbb{E}_x \left[ h_k(x) \right] \right], \tag{11}$$

$$\text{Variance of IGD: } \mathbb{V}_{G_1} \left[ \mathbb{E}_x \left[ 2(\bar{h}_{\mathcal{G}_1}(x) - \bar{h}_{\mathcal{G}_2}(x)) \right] \right] = \mathbb{V}_{G_1} \left[ \mathbb{E}_x \left[ 2(2\bar{h}_{\mathcal{G}_1}(x) - 2\bar{h}(x)) \right] \right]. \tag{12}$$

Then,

$$\text{Variance of IGD} = 16 \mathbb{V}_{G_1} \left[ \mathbb{E}_x \left[ \bar{h}_{\mathcal{G}_1}(x) \right] \right] \qquad\qquad \text{since } \mathbb{E}_x \bar{h}(x) \text{ is a constant w.r.t. } \mathcal{G}_1 \tag{13}$$

$$= 16 \mathbb{V}_{G_1} \left[ \frac{1}{M} \sum_{i \in \mathcal{G}_1} \mathbb{E}_x \left[ h_i(x) \right] \right] \qquad\qquad \text{by linearity of expectation} \tag{14}$$

$$= 16 \cdot \frac{2M - M}{2M - 1} \frac{\mathbb{V}_k \left[ \mathbb{E}_x \left[ h_k(x) \right] \right]}{M} \qquad\qquad \text{variance of sample mean without replacement} \tag{15}$$

$$\approx \frac{8}{M} \mathbb{V}_k \left[ \mathbb{E}_x \left[ h_k(x) \right] \right] \tag{16}$$

$$= \frac{1}{M} \text{Variance of IDD.} \tag{17}$$

**Summary.** Under uniform weights, IGD underestimates the normalized full-pairwise objective, but enjoys reduced variance in the order of $1/M$ compared to IDD. We emphasize that this is not a claim about how GALA works in practice, but a useful theoretical anchor that gives an exact characterization of the variance reduction mechanism that is understood intuitively.

### B.2. Empirical Bias and Variance of IGD under GALA Weights

To support our theoretical analysis, we empirically investigate the bias and variance of IGD under adaptive GALA weights. Figure 6 compares $\mathcal{L}_{\text{full}}$, $\mathcal{L}_{\text{IGD}}$, and $\mathcal{L}_{\text{IDD}}$ (subplot (a)), together with their variances (subplots (b) and (c)), after 100 rounds of standard GALA training ($\tau = 1.0$). Starting from four source domains with SVHN and MNIST-M as targets, additional Digit-18 domains are incrementally added in order of increasing task difficulty (i.e., lowest self-performance first; see

Appendix E.1.3). The 17-source setting corresponds to the complete Digit-18 benchmark. IDD variance is computed as the variance of discrepancies across all combinations of fine-tuned source classifiers using the shared $G'$. For the 4-, 6-, and 8-source settings, IGD is computed by aggregating classifiers within all possible group splits, measuring discrepancies across splits via $G'$, and subsequently computing the mean and variance. For the 17-source setting, the IGD mean and variance are approximated via random sampling over 500 splits into two groups.

The results confirm that IGD is biased and consistently underestimates $\mathcal{L}_{\text{full}}$. However, this bias does not appear detrimental in practice. Across all evaluated benchmarks, GALA consistently improves both stability and accuracy, particularly as the number of source domains increases, with no indication of the underfitting behavior that would be expected from a strongly biased estimator. In contrast, the variance is substantially reduced compared to IDD. Notably, the gap in variance increases as the number of source domains increases, highlighting the advantage of IGD over IDD in large-scale settings.

To further analyze the relationship between $\mathcal{L}_{\text{full}}$ and $\mathcal{L}_{\text{IGD}}$, Figure 7 presents a convergence analysis of IGD compared to $\mathcal{L}_{\text{full}}$ on Digit-Five. The results are obtained using standard GALA training for 300 rounds with $\tau = 1.0$. Every 10 rounds, $\mathcal{L}_{\text{full}}$ is computed as the average discrepancy across all possible combinations of fine-tuned source classifiers using the shared $G'$. IGD is computed by averaging classifiers within each group, measuring discrepancies across all possible group splits through $G'$, and then averaging the resulting discrepancies.

Although IGD systematically underestimates $\mathcal{L}_{\text{full}}$, its optimization dynamics closely follow those of $\mathcal{L}_{\text{full}}$ throughout training. While these experiments directly optimize $\mathcal{L}_{\text{IGD}}$, monitoring $\mathcal{L}_{\text{full}}$ reveals that the full objective is simultaneously optimized. This observation suggests that IGD provides an effective and practical approximation of $\mathcal{L}_{\text{full}}$, which is otherwise computationally infeasible to evaluate exactly.

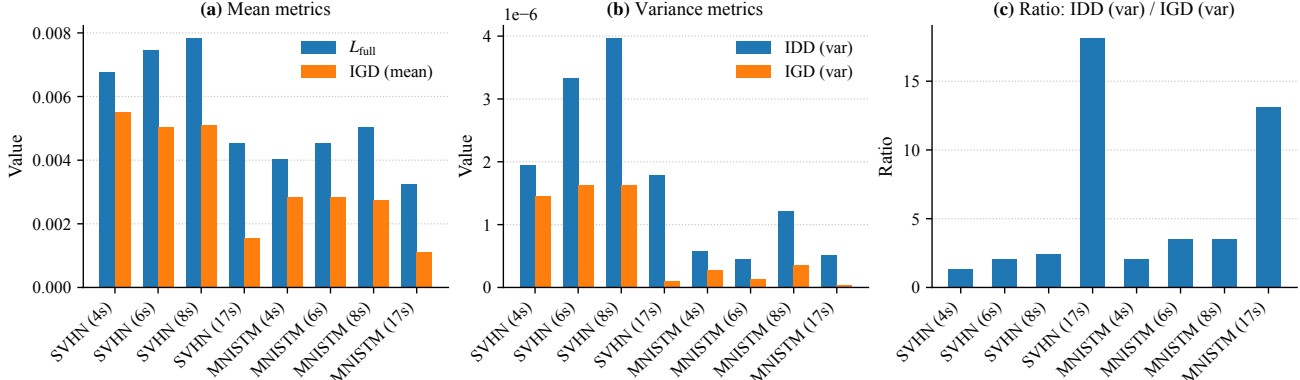

Figure 6. Bias and variance analysis of $\mathcal{L}_{\text{full}}$, $\mathcal{L}_{\text{IGD}}$ and $\mathcal{L}_{\text{IDD}}$ on Digit-Five and Digit-18 (mean over three runs).

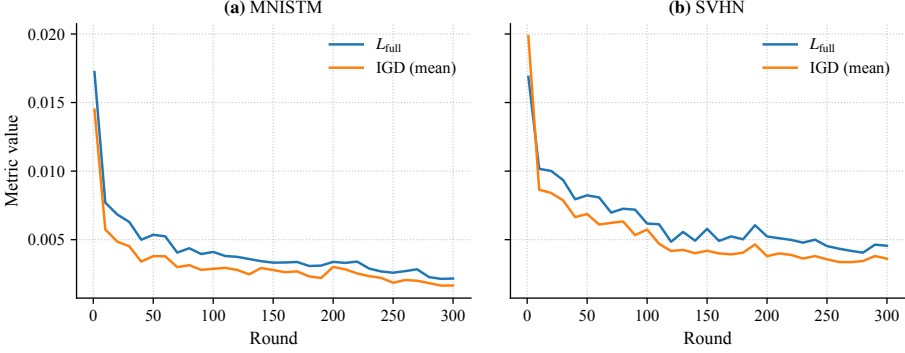

Figure 7. Convergence analysis of IGD compared to $\mathcal{L}_{\text{full}}$ on Digit-Five (4 sources): **(a)** MNIST-M as target and **(b)** SVHN as target. Results are obtained using standard GALA training for 300 rounds with $\tau = 1.0$. Every 10 rounds, $\mathcal{L}_{\text{full}}$ is computed as the average discrepancy across all possible combinations of fine-tuned source classifiers using the shared $G'$. IGD is computed by averaging classifiers within each group, measuring discrepancies across all possible splits through $G'$, and then averaging these discrepancies.

## C. GALA: Algorithm Overview

We next describe the proposed GALA framework. The complete procedure is detailed in Algorithm 1.Each round begins with the server broadcasting the global model $(G_t, F_t)$ to all domains. Domains compute class-wise centroids and upload them to the server, which calculates normalized relevance scores via MDMGB+ to weight each source's contribution. Sources update their models locally using cross-entropy loss. The server aggregates feature extractors with similarity-based weights to form a shared extractor $G'$, sent back to sources. With $G'$ frozen, sources fine-tune predictors and return updates to the server. The server randomly partitions predictors into two groups, averages them, and sends both groups with $G'$ to the target. The target updates $G'$ to $G''$ by minimizing IGD loss between group predictions. The server merges the group predictors into a global predictor $F_{t+1}$ and sets $G_{t+1} \leftarrow G''$, updating the model as $h_{t+1} = F_{t+1} \circ G_{t+1}$. This completes one round.

---

**Algorithm 1** Training Process of GALA

---

**Input:** Source datasets $\{\mathbb{D}_S^n\}_{n=1}^N$, target dataset $\mathbb{D}_T$, initial model $(G, F)$, total rounds $T$, temperature $\tau$

**for** $t = 1$ **to** $T$ **do**

  Broadcast global model $(G_t, F_t)$ to all domains

  **for** each class $c \in C$ **do**

    **in parallel for each source** $n$**:**

    Compute $r_n^c \leftarrow \frac{\sum_{x \in \mathbb{D}_S^n} \delta_c(x) G(x)}{\sum_{x \in \mathbb{D}_S^n} \delta_c(x)}$

    Target computes $r_T^c \leftarrow \frac{\sum_{x \in \mathbb{D}_T} \delta_c(x) G(x)}{\sum_{x \in \mathbb{D}_T} \delta_c(x)}$

  **end for**

  Compute domain similarity $S(r_T, r_n)$ via Eq. 6

  $w_n \leftarrow \texttt{MDMGBPlus}(r_T, r_n)$ via Eq. 7

  **for** each source $n$ **in parallel do**

    Initialize $(G_n, F_n) \leftarrow (G_t, F_t)$

    Update $(G_n, F_n)$ by optimizing $\mathbb{E}_{(x,y) \sim \mathbb{D}_S^n}[\ell(F_n(G_n(x)), y)]$

  **end for**

  Aggregate $G' \leftarrow \sum_n w_n G_n$ and broadcast to sources

  **for** each source $n$ **in parallel do**

    Freeze $G'$, fine-tune $F_n$ on $\mathbb{D}_S^n$

    Send updated $F_n$ to server

  **end for**

  Randomly split sources into groups $\mathcal{G}_1$ and $\mathcal{G}_2$

  **for** each group $\mathcal{G}_i \in \{\mathcal{G}_1, \mathcal{G}_2\}$ **do**

    **for** each source $n \in \mathcal{G}_i$ **do**

      Compute normalized weight $\tilde{w}_n$ via Eq. 8

    **end for**

    $F_{\mathcal{G}_i} \leftarrow \sum_{n \in \mathcal{G}_i} \tilde{w}_n F_n$

    $w_{\mathcal{G}_i} \leftarrow \sum_{n \in \mathcal{G}_i} w_n$

  **end for**

  Send $(F_{\mathcal{G}_1}, F_{\mathcal{G}_2}, G')$ to target domain

  Target updates $G'$ to $G''$ by minimizing $\mathcal{L}_{\text{IGD}}$

  $G_{t+1} \leftarrow G''$

  Aggregate global predictor $F_{t+1} \leftarrow w_{\mathcal{G}_1} F_{\mathcal{G}_1} + w_{\mathcal{G}_2} F_{\mathcal{G}_2}$

**end for**

---

## D. Implementation Details

We provide here the complete architectural specifications and training hyperparameters for reproducibility. The code for reproducing the experiments, as well as the Digit-18 benchmark, is available on GitHub[4].

---

[4]github.com/mdppml/GALA.git

## D.1. Architectures

For experiments on Digit-Five and Digit-18, we use a lightweight 2-layer CNN as the feature extractor, followed by a 3-layer MLP predictor. The full architecture is detailed in Table 4. For Office-Caltech10 and DomainNet, we adopt a ResNet101 backbone pretrained on ImageNet, followed by a task-specific MLP predictor as outlined in Tables 5 and 6.

*Table 4.* Digit datasets Model Architecture.

| Layer | Output Size | Kernel / Units | Details |
|---|---|---|---|
| Input | $3 \times 32 \times 32$ | - | RGB Image |
| Conv2D + BN + ReLU | $64 \times 32 \times 32$ | $5 \times 5$ | padding=2 |
| MaxPool2D | $64 \times 16 \times 16$ | $3 \times 3$ | stride=2, padding=1 |
| Conv2D + BN + ReLU | $128 \times 16 \times 16$ | $5 \times 5$ | padding=2 |
| MaxPool2D | $128 \times 8 \times 8$ | $3 \times 3$ | stride=2, padding=1 |
| Flatten | 8192 | - | |
| Dropout + FC + BN + ReLU | 3072 | - | p=0.5 |
| Dropout + FC + BN + ReLU | 100 | - | p=0.5 |
| Dropout + FC + BN + Softmax | 10 | - | p=0.5 |

*Table 5.* ResNet-based Predictor Architecture for Office-Caltech10.

| Layer | Output Size | Units | Details |
|---|---|---|---|
| ResNet101 Backbone | 1000 | - | Pretrained on ImageNet |
| Dropout + FC + BN + ReLU | 500 | - | p=0.5 |
| FC + BN + Softmax | $\{10, \text{Number of Classes}\}$ | - | Task-specific classes |

*Table 6.* ResNet-based Predictor Architecture for DomainNet.

| Layer | Output Size | Units | Details |
|---|---|---|---|
| ResNet101 Backbone | 2048 | - | Pretrained on ImageNet |
| FC | $\{345, \text{Number of Classes}\}$ | - | Task-specific classes |

## D.2. Training Details

Table 7 summarizes the training parameters used for each benchmark. All models are trained using SGD with momentum 0.9 and weight decay $5 \times 10^{-4}$. Communication occurs once per round ($r = 1$), and each training phase (source training, fine-tuning, adversarial alignment) is performed for one epoch. Following (Feng et al., 2021), we apply mixup augmentation ($\alpha = 0.2$) for Office-Caltech10 only. The temperature parameter $\tau$ is set to 0.03 for Clipart and Sketch targets and 0.05 for all other DomainNet targets, 1.0 for Office-Caltech10 and Digit-18, and 0.2 for Digit-Five.

*Table 7.* Implementation details of our GALA on four benchmark datasets.

| Parameters | Digit-Five | Digit-18 | Office-Caltech10 | DomainNet |
|---|---|---|---|---|
| Data Augmentation | None | | Mixup ($\alpha = 0.2$) | None |
| Backbone | 2-layer CNN | | ResNet101 (pretrained=True) | |
| Optimizer | SGD with momentum = 0.9 and weight decay =$5 \times 10^{-4}$ | | | |
| Learning Rate Schedule | CustomLR ($\gamma$=0.75) | | ExponentialLR ($\gamma$=0.9) | CustomLR ($\gamma$=0.75) |
| Batch Size | 128 | | | 50 |
| Total Rounds | 500 | | | 20 |
| Communication Rounds | $r = 1$ | | | |
| Temperature | 0.2 | 1.0 | | 0.03–0.05 |

# E. Datasets

**DomainNet.** (Peng et al., 2019b) A widely used domain adaptation benchmark with 345 classes across six domains, designed to test highly heterogeneous domain shifts.

**Office-Caltech10.** Office-Caltech10 consists of for domains: Amazon, Webcam, DSLR and Caltech. The images show objects from 10 different classes which are shared between Office (Saenko et al., 2010) and Caltech-265 (Griffin et al., 2022) datasets.

**Digit-Five.** The Digit-Five dataset (Zhao et al., 2020) is a popular benchmark for digit recognition. It consists of the following five datasets, each representing a separate domain: MNIST, MNIST-M, Street-View House Numbers (SVHN), Synthetic Digits (SYN), and USPS.

### E.1. Digit-18 Benchmark

**Digit-18** is our proposed large-scale benchmark composed of 18 domains, created by applying systematic transformations to existing digit datasets. It is specifically designed to evaluate the robustness and scalability of UMDA methods in high-source scenarios. Our goal was to ensure sufficient variability and domain shifts across the domains. Thus, we did not apply each transformation to every dataset, as some domains are already similar.

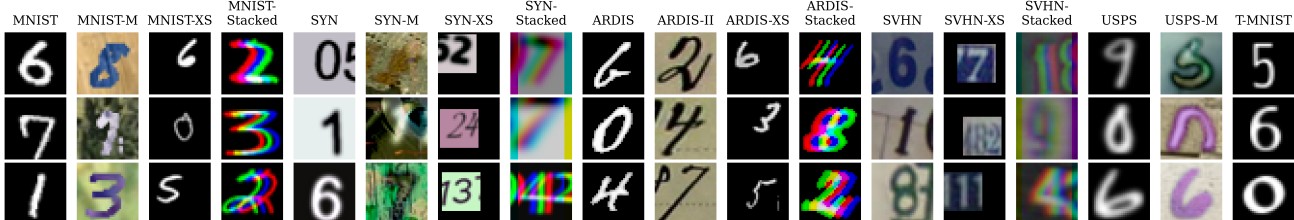

*Figure 8.* Sample images from each domain in the **Digit-18** dataset.

#### E.1.1. BASE DATASETS

- **ARDIS** (Kusetogullari et al., 2019): A historical handwritten digit dataset extracted from Swedish church records. We use 6,600 training and 1,000 testing samples. We include two variants: a normalized version (matched to MNIST) and an unprocessed version with original grayscale backgrounds and image noise, referred to as **ARDIS II**.

- **TMNIST** (Magre & Brown, 2022): Typography-MNIST contains 22,400 training and 7,500 test images of digits rendered in various fonts. The images are grayscale on a black background, similar to MNIST but with greater stylistic diversity.

#### E.1.2. DOMAIN TRANSFORMATIONS

We applied the following transformation strategies to simulate diverse and challenging domain shifts:

- **Background Augmentation:** Following MNIST-M (Ganin et al., 2016), we overlay complex colored backgrounds on digit images from SYN, SVHN, and USPS to create SYNM, SVHNM, and USPSM.

- **Scaling:** Original digit images are resized to $20 \times 20$ and re-centered on a $32 \times 32$ black canvas. Applied to MNIST, SYN, SVHN, and ARDIS, yielding *-XS domains (e.g., MNISTXS).

- **Stacking:** We introduce pixel-level channel misalignments by shifting R, G, B channels in opposite directions. Applied to grayscale domains, this generates color interference effects. Used for MNIST, SYN, SVHN, and ARDIS to generate *-STACK domains.

#### E.1.3. DOMAIN SHIFT ANALYSIS

To assess domain similarity and difficulty, we trained simple models on each domain independently and evaluated them across all other domains. These models used the same architecture and training settings as in the UFDA experiments

(500 epochs, SGD with momentum 0.9, fixed learning rate 0.001). The accuracy matrix in Figure 9 reveals cross-domain generalization trends and helps characterize inter-domain shifts.

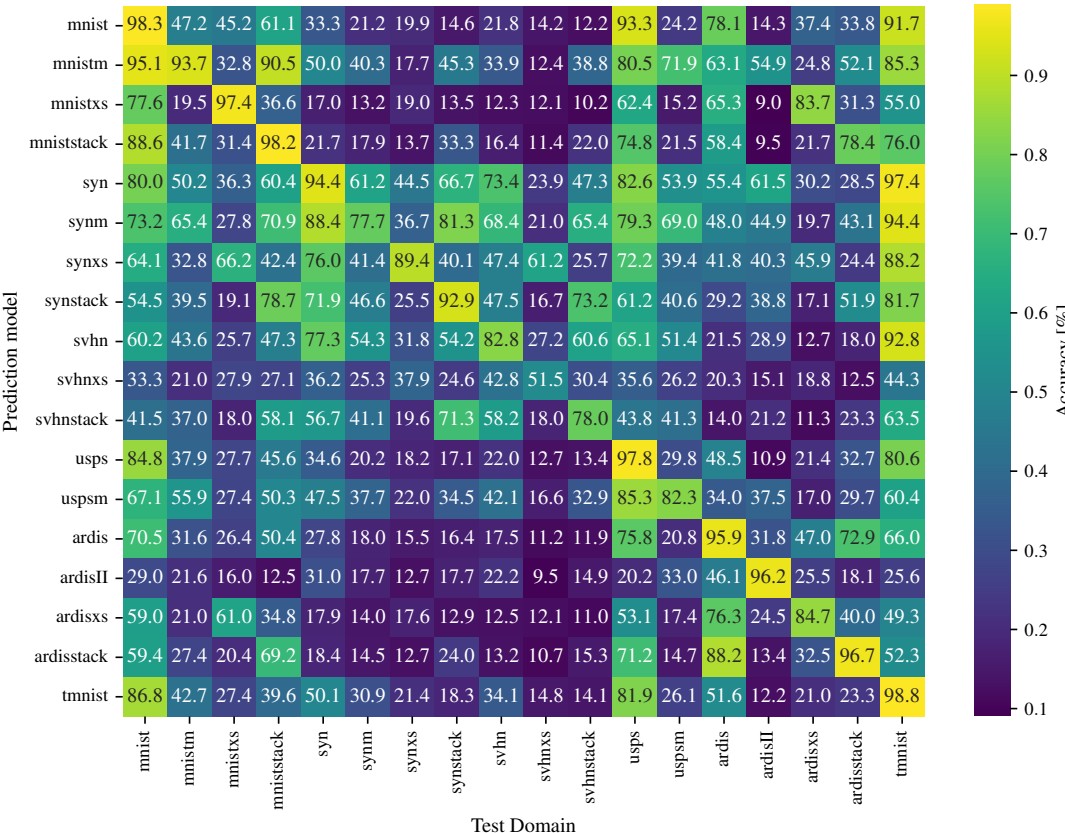

*Figure 9.* Cross-domain similarity matrix: each row corresponds to a model trained on a source domain and evaluated on all target domains.

Notably, models trained on clean datasets (e.g., MNIST) fail to generalize well to complex variants (e.g., SYNM), while models trained on background-augmented domains (e.g., MNISTM) transfer better to simpler settings. These insights informed the domain selection process and help contextualize results in our experiments.

## F. Results on Digit-Five and Office-Caltech10

**Digit-Five.** As shown in Table 8, GALA achieves the highest average accuracy on Digit-Five and performs best on all but two target domains. On the remaining targets, it is only marginally below the best method with differences of around 0.2%. These results demonstrate that GALA matches or surpasses state-of-the-art methods on this standard benchmark, serving as a strong baseline for the more challenging settings considered next.

**Office-Caltech10.** On this small benchmark, GALA ranks second among distributed methods with 97.8% accuracy, closely matching KD3A (97.9%). While KD3A is slightly better on Amazon and Caltech, GALA achieves the highest accuracy on DSLR and Webcam, including 100% on DSLR.

## G. Communication-cost analysis

In this work, we focus on the algorithmic scalability and stability of the distributed UMDA objective under many sources. Practical FL networking issues (client dropouts, bandwidth limits, etc.) are independent challenges, and we follow the same standard assumptions used in most FL literature, where all clients participate in each round.

*Table 8.* UMDA accuracy (%) on the Digit-Five dataset.

| Setting | Method | *mnist* | *mnistm* | *svhn* | *syn* | *usps* | Avg |
|---|---|---|---|---|---|---|---|
| W/o DA | Oracle | $99.5_{\pm 0.08}$ | $95.4_{\pm 0.15}$ | $92.3_{\pm 0.14}$ | $98.7_{\pm 0.04}$ | $99.2_{\pm 0.09}$ | 97.0 |
| | Source-only | $92.3_{\pm 0.91}$ | $63.7_{\pm 0.83}$ | $71.5_{\pm 0.75}$ | $83.4_{\pm 0.79}$ | $90.71_{\pm 0.54}$ | 80.3 |
| Centralized | MDAN | $97.2_{\pm 0.98}$ | $75.7_{\pm 0.83}$ | $82.2_{\pm 0.82}$ | $85.2_{\pm 0.58}$ | $93.3_{\pm 0.48}$ | 86.7 |
| | $M^3$SDA | $98.4_{\pm 0.68}$ | $72.8_{\pm 1.13}$ | $81.3_{\pm 0.86}$ | $89.6_{\pm 0.56}$ | $96.2_{\pm 0.81}$ | 87.7 |
| | CMSS | $99.0_{\pm 0.08}$ | $75.3_{\pm 0.57}$ | $88.4_{\pm 0.54}$ | $93.7_{\pm 0.21}$ | $97.7_{\pm 0.13}$ | 90.8 |
| | DSBN | 97.2 | 71.6 | 77.9 | 88.7 | 96.1 | 86.3 |
| Distributed | FADA | $91.4_{\pm 0.7}$ | $62.5_{\pm 0.7}$ | $50.5_{\pm 0.3}$ | $71.8_{\pm 0.5}$ | $91.7_{\pm 1}$ | 73.6 |
| | SHOT | $98.2_{\pm 0.37}$ | $80.2_{\pm 0.41}$ | $84.5_{\pm 0.32}$ | $91.1_{\pm 0.23}$ | $97.1_{\pm 0.28}$ | 90.2 |
| | SFDA | 99.1 | 72.3 | 86.0 | 90.4 | 98.1 | 89.2 |
| | KD3A | $99.2_{\pm 0.12}$ | $87.3_{\pm 0.23}$ | $85.6_{\pm 0.17}$ | $89.4_{\pm 0.28}$ | $\mathbf{98.5_{\pm 0.25}}$ | 92.0 |
| | FACT | $\mathbf{99.3_{\pm 0.12}}$ | $91.4_{\pm 0.53}$ | $90.9_{\pm 0.40}$ | $94.8_{\pm 0.22}$ | $98.3_{\pm 0.11}$ | 95.0 |
| | GALA | $99.2_{\pm 0.05}$ | $\mathbf{93.0_{\pm 0.43}}$ | $91.2_{\pm 0.16}$ | $\mathbf{95.2_{\pm 0.17}}$ | $98.3_{\pm 0.10}$ | **95.4** |

*Table 9.* UMDA accuracy (%) on the Office-Caltech10. [‡]: With data augmentation.

| Methods | *amazon* | *caltech* | *dslr* | *webcam* | Avg |
|---|---|---|---|---|---|
| Oracle | 99.7 | 98.4 | 99.8 | 99.7 | 99.4 |
| Source-only | 86.1 | 87.8 | 98.3 | 99.0 | 92.8 |
| MDAN | 98.9 | 98.6 | 91.8 | 95.4 | 96.1 |
| M$^3$SDA | 94.5 | 92.2 | 99.2 | 99.5 | 96.4 |
| CMSS | 96.0 | 93.7 | 99.3 | 99.6 | 97.2 |
| DSBN | 93.2 | 91.6 | 98.9 | 99.3 | 95.8 |
| DANE | 97.4 | 97.3 | 100.0 | 100.0 | 98.7 |
| FADA | $84.2_{\pm 0.5}$ | $88.7_{\pm 0.5}$ | $87.1_{\pm 0.6}$ | $88.1_{\pm 0.4}$ | 87.1 |
| SHOT | 96.4 | 96.2 | 98.5 | 99.7 | 97.7 |
| FACT | 96.3 | 95.5 | 99.4 | 99.0 | 97.6 |
| KD3A[‡] | $\mathbf{97.4_{\pm 0.08}}$ | $\mathbf{96.4_{\pm 0.11}}$ | $98.4_{\pm 0.08}$ | $99.7_{\pm 0.02}$ | **97.9** |
| GALA[‡] | $96.5_{\pm 0.19}$ | $95.0_{\pm 0.17}$ | $\mathbf{100.0_{\pm 0.00}}$ | $99.8_{\pm 0.17}$ | 97.8 |

Importantly, GALA does not introduce heavy additional communication. Each source sends only a centroid vector for each class $C$, which is negligible in size. We also performed an additional communication-cost analysis. The total upload/download cost per round (approx.) is reported in Table 10.

*Table 10.* Total upload/download cost per round (approx.).

| Method | 5 Sources | | | 10 Sources | | |
|---|---|---|---|---|---|---|
| | Source | Server | Target | Source | Server | Target |
| FACT | $\sim$ 205 MB | $\sim$ 618 MB | $\sim$ 205 MB | $\sim$ 205 MB | $\sim$ 618 MB | $\sim$ 205.63 MB |
| GALA | $\sim$ 207 MB | $\sim$ 1.24 GB | $\sim$ 205 MB | $\sim$ 207 MB | $\sim$ 2.27 GB | $\sim$ 205.63 MB |
| KD3A | $\sim$ 205 MB | N/A | $\sim$ 1.03 GB | $\sim$ 205 MB | N/A | $\sim$ 2.05 GB |

Overall, FACT is the most communication-efficient due to its selective strategy. Although GALA involves all sources every round, its total communication cost remains comparable to KD3A while providing a centralized FL workflow and not requiring the target to receive all source models.

## H. GALA with Partial Client Participation

GALA naturally supports variants with partial client participation, enabling reduced computational cost and robustness to client unavailability. To evaluate this setting, we conducted additional experiments on Digit-18 in which only a random

subset of clients participates in each communication round. The results are compared with FACT in Table 11. We report the mean and standard deviation over three independent runs.

*Table 11.* GALA performance under different client participation rates on Digit-18 dataset.

| Target | *mnist* | *svhn-xs* |
|---|---|---|
| FACT | $87.03 \pm 1.47$ | $80.57 \pm 4.90$ |
| GALA (Full participation) | $95.17 \pm 0.05$ | $88.47 \pm 0.05$ |
| GALA (75% participation) | $94.73 \pm 0.25$ | $88.67 \pm 0.29$ |
| GALA (50% participation) | $94.00 \pm 0.33$ | $87.93 \pm 0.12$ |
| GALA (25% participation) | $92.40 \pm 0.64$ | $84.20 \pm 1.92$ |

The results show that reducing client participation gradually decreases performance, although the drop in accuracy from full participation to 25% participation remains relatively small. Importantly, GALA remains stable and consistently outperforms FACT across all participation levels. These findings suggest that GALA can be effectively adapted to different computational budgets by selectively excluding clients. However, partial participation does not substantially reduce total training time due to the high degree of parallelization (see Section 4.3), and the resulting runtime reduction is comparable to that obtained by decreasing the total number of clients (Table 3).

We further observe that the learned client weights stabilize after several communication rounds, as illustrated in Figure 12. This observation suggests a potential extension in which clients with consistently low weights could be excluded during later stages of training, potentially leading to additional efficiency gains.

## I. Robustness Analysis of FACT

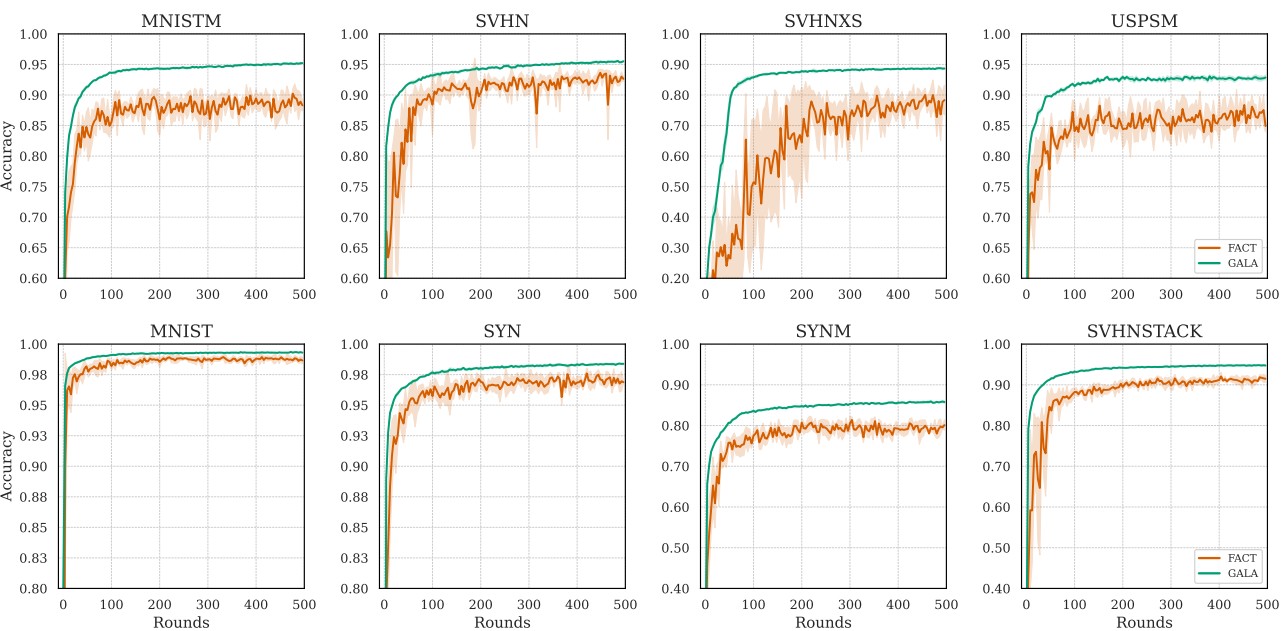

*Figure 10.* Test accuracy over training rounds for different target domains in the full Digit-18 setup (extension of Figure 3).

FACT randomly selects two source domains in each communication round to perform inter-domain distance minimization. This random pairing strategy introduces instability during training, as model updates become highly sensitive to the particular source combination selected in each round. As a result, we observe frequent fluctuations in test accuracy, especially on more challenging target domains, as illustrated in Figure 10. To examine this behavior in more detail, we analyze training dynamics on the Digit-18 benchmark. Figure 11 shows round-to-round changes in test accuracy for two particularly challenging target domains, SVHNXS and MNISTM. For interpretability, we annotate the source domain pairs selected in rounds exhibiting the largest single-round accuracy increases and decreases. To focus on convergence behavior, we restrict

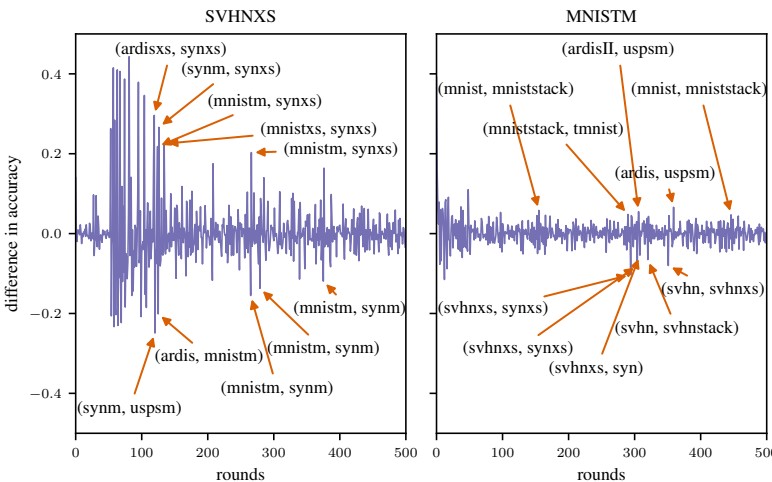

*Figure 11.* Round-to-round accuracy difference of FACT. Clients associated with the highest single-round accuracy increases and decreases are annotated.

this analysis to the phase after the first 110 communication rounds (i.e., after the warm-up period).

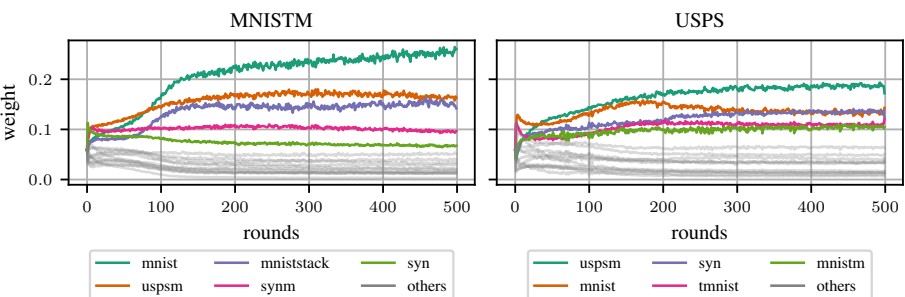

*Figure 12.* Evolution of source weights assigned by MDMGB+ for MNISTM and USPS (Digit-18 setting). The five most highly weighted source domains are highlighted. Shifts over time reflect both domain similarity and the model's adaptation to target-specific learning needs.

For SVHNXS, the largest accuracy drops occur when both selected sources are *-M domains, all of which exhibit similarity scores below 25% with SVHNXS (see similarity matrix in Figure 9). Since SVHNXS consists of black-and-white digit images with extensive black backgrounds, it cannot effectively leverage the colorful backgrounds characteristic of *-M sources, leading to negative transfer. In contrast, pairings such as MNISTM with SYNXS result in marked performance improvements, indicating that a dissimilar source can still be beneficial when combined with a complementary one. Several *-XS sources, particularly SYNXS, consistently produce significant accuracy gains. This behavior can be attributed to similar data generation processes across these domains, which enable the model to better capture the characteristics of SVHNXS. Notably, SYNXS, the most similar source to SVHNXS, appears in all beneficial source pairs. We further observe that SYNXS is consistently assigned the highest weight by GALA, demonstrating our method's ability to identify and emphasize the most relevant sources.

A similar pattern emerges for MNISTM: selecting MNIST-like and -M domains typically leads to improvements in test accuracy relative to the previous round, whereas selecting SVHN- domains and SYNXS, which have similarity scores below 50%, often results in substantial accuracy drops.

Overall, these results highlight the robustness limitations of FACT's random source selection mechanism in large-scale multi-source settings.

## J. Adaptive Source Weighting in GALA

GALA dynamically assigns a weight to each source client in every training round, determining its influence on the shared model. Figure 12 illustrates the evolution of these weights for the target domains MNISTM and USPS in the Digit-18 setting. The top five most highly weighted sources are highlighted.

For MNISTM, the top-weighted domains align with those found most similar in our similarity analysis. These include MNIST-like and *-M datasets, supporting the expectation that their combination is well-suited for learning MNISTM. The USPS plot illustrates the value of dynamic re-weighting. In early rounds, simpler domains like MNIST and TMNIST dominate. Later, the weights shift toward USPSM and SYN, reflecting a focus on learning finer details and USPS-specific features.

