# OpenReview forum: "Scaling Unsupervised Multi-Source Federated Domain Adaptation through Group-Wise Discrepancy Minimization"
_ICML.cc/2026/Conference — ICML 2026 regular_

### Official Review · Reviewer_FFe9 · 2026-03-04

**Soundness:** 3
**Presentation:** 3
**Significance:** 2
**Originality:** 2
**Overall Recommendation:** 4
**Confidence:** 3

**Summary:**

This paper targets multi-source domain adaptation, concretely, the federated one, where we do not have access to source data. Concretely, they propose a method called Grouping-based Adaptive Learning, GALA, to enable multi-source domain adaptation with large amount of source domain. GALA has a inter-group objective which quantifies pairwise discrepancy, it is in linear complexity, without computing the conventional pairwise discrepancy. In addition, a temperature-scaled centroid-based weighting scheme facilitates performance. They also propose a new dataset called digit-18, which has 18 domains. Extensive experiment results prove the effectiveness of the proposed method. It also enjoys stability and robustness over previous methods.

**Compliance With Llm Reviewing Policy:**

Affirmed.

**Final Justification:**

After reading the rebuttal, I keep my score

**Key Questions For Authors:**

1. See weaknesses. My key problem is on the complexity of the dataset.

**Limitations:**

yes.

**Strengths And Weaknesses:**

Strengths:
1. I think this paper is well-organized and easy to read. The authors introduce the problem, theoretical insights, method design, and experiment results in a clear manner.
2. The experiment section covers recent baseline methods, as well as runtime analysis and parameter analysis.
3. It is good to not only pursue performance, but also consider the stability and robustness of the method.
4. A dataset is proposed for this problem. The discussions on the limitations hit my box. I think it will be promising to develop a more large-scale benchmark.

Weakness:
1. I think the current dataset in the experiment section appear to be a little bit simple. Perhaps the authors can consider some more complicated datasets (even do not have so many domains). It will be acceptable if the proposed method does not perform well. Failure cases can also help us understand the work.

---

> ### Author Rebuttal · Authors · 2026-03-30
>
> Thank you for the positive assessment and helpful suggestion.
>
> We would like to clarify our dataset choice. Our work targets federated domain adaptation in a many-source setting, which is not covered by existing benchmarks. Most standard datasets used in this literature, such as Digit-5, Office-Caltech10, and DomainNet, contain only 4–6 domains. This limitation motivated us to introduce Digit-18, which extends Digit-5 to a substantially larger number of domains and enables controlled evaluation of scalability in this setting. While digit datasets are visually simpler, Digit-18 includes several more challenging domains, such as SVHNXS, SYNM, and SVHNSTACK.
>
> In Appendix Fig. 7, we provide a cross-domain similarity matrix based on centralized supervised training across all domains, which highlights the non-trivial shifts and diversity within Digit-18. We agree that future work should explore more complex many-domain benchmarks, and we view Digit-18 as a first step in that direction.
>
> At the same time, we agree that validation on a more complex dataset is important. Following the reviewer’s suggestion, we conducted additional experiments on DomainNet, a significantly more challenging benchmark with 345 classes.
>
> | |Clipart|Infograph|Painting|Quickdraw|Real|Sketch|Avg|
> |-|-|-|-|-|-|-|-|
> |KD3A*|69.7|**21.2**|**58.8**|15.1|**70.4**|57.9|48.85|
> |GALA|**72.0**|20.8|56.4|**20.2**|68.9|**58.4**|**49.45**|
>
> *without data augmentation
>
> For simplicity, we did not use any data augmentation. Although DomainNet has relatively few domains (6) and is not our primary target setting, GALA remains competitive and achieves better average performance than KD3A. These results support that GALA extends beyond digit benchmarks while maintaining its advantages. We will include a full comparison table on DomainNet in the camera-ready manuscript.
>
> We believe this additional experiment directly addresses the concern about dataset simplicity and further supports the broader applicability of our method.

---

> > ### Author Rebuttal · Reviewer_FFe9 · 2026-04-06
> >
> > Thank you for your rebuttal. After reading it, I still want to accept this paper.

---

> > > ### Author Response · Authors · 2026-04-06
> > >
> > > Thank you for the positive evaluation and for acknowledging our rebuttal. We appreciate your time and thoughtful feedback, we are glad the rebuttal addressed your concerns regarding dataset complexity.

---

### Official Review · Reviewer_xgZA · 2026-03-09

**Soundness:** 4
**Presentation:** 3
**Significance:** 3
**Originality:** 4
**Overall Recommendation:** 5
**Confidence:** 4

**Summary:**

This paper proposes GALA, a scalable and robust federated Unsupervised Multi-source Domain Adaptation framework designed to address the high computational overhead and training instability of existing methods as the number of source domains increases. GALA employs a novel inter-group discrepancy minimization objective to approximate global alignment with linear complexity, coupled with a centroid-based temperature-weighting strategy that dynamically prioritizes source domains relevant to the target. To evaluate performance in high-diversity scenarios, the authors introduce Digit-18, a new benchmark comprising 18 datasets. Experiments demonstrate that GALA achieves state-of-the-art results on standard benchmarks and significantly outperforms prior methods in large-scale, high-diversity settings, ensuring stable convergence and efficient scalability.

**Compliance With Llm Reviewing Policy:**

Affirmed.

**Final Justification:**

I appreciate the authors' detailed responses and finalize to accept.

**Key Questions For Authors:**

No

**Limitations:**

Yes

**Strengths And Weaknesses:**

Strengths:

1.	Exceptional Scalability: GALA’s inter-group discrepancy minimization reduces complexity from quadratic to linear, enabling efficient handling of hundreds of heterogeneous sources where existing methods fail or become infeasible.

2.	Robustness via Dynamic Weighting: A centroid-based temperature-scaled strategy dynamically prioritizes source domains aligned with the target, mitigating negative transfer and enhancing stability under complex distribution shifts.

3.	Novel High-Diversity Benchmark: The new Digit-18 benchmark, featuring 18 diverse datasets, overcomes current limitations by providing a standardized platform for evaluating performance in large-scale, highly heterogeneous real-world scenarios.

Weakness:

1.	The Figure is unclear: The figure 1 places all figures and models in a box, causing it is difficult to distinguish boundaries between the clients. Besides, it is also unclear which steps are completed in the server.

2.	The organization is unclear. The paper puts the motivation and prior work in the same section, i.e., methodology section, causing it is hard to distinguish the proposed method from the whole section.

3.	It will be better if the specific and explicit theoretical improvement can be given instead of the theoretical intuition.

4.	NonIID is also a serious problem the federated settings among different domains. The paper should also discuss whether some NonIID methods can also solve the target problem, e.g., [1,2,3].

[1] FedCDA: Federated Learning with Cross-rounds Divergence-aware Aggregation. In The Twelfth International Conference on Learning Representations (ICLR 2024), Vienna, Austria, May 7th to May 11th, 2024.

[2] DaFKD: Domain-aware Federated Knowledge Distillation. The Thirty-Fourth IEEE/CVF Conference on Computer Vision and Pattern Recognition (CVPR 2023), Vancouver, Canada, June 18-22, 2023.

[3] FedNLR: Federated Learning with Neuron-wise Learning Rates. The 30th SIGKDD Conference on Knowledge Discovery and Data Mining (KDD 2024), Barcelona, Spain, 25-29 August 2024.

---

> ### Author Rebuttal · Authors · 2026-03-30
>
> Thank you for the positive evaluation and for the helpful suggestions. We appreciate the reviewer’s comments on presentation and related work.
>
> Regarding the suggested comparisons with FedCDA and FedNLR, these methods primarily address non-IID label-skew heterogeneity in supervised federated learning. In contrast, our work targets federated unsupervised multi-source domain adaptation, where the main challenge is cross-domain feature shift with an unlabeled target and multiple source domains. Because the problem formulation, supervision regime, and adaptation objective are different, these methods are not directly comparable to GALA.
>
> DaFKD is relevant to the broader non-IID federated learning literature. However, it focuses on supervised federated distillation under client heterogeneity, whereas GALA addresses federated unsupervised multi-source domain adaptation with an unlabeled target. We will clarify this distinction and cite DaFKD as related work rather than as a direct baseline.
>
> That said, we agree that the interaction between domain shift and label skewness is an important and practically relevant direction. Extending multi-source domain adaptation methods to jointly handle both forms of heterogeneity would be an interesting topic for future work. We will clarify this distinction in the revised manuscript and add a brief discussion of how GALA may be combined with these non-IID federated optimization or distillation approaches in more general settings.
>
> We also appreciate the comments on presentation. We will revise Fig. 1 to better distinguish client and server operations, and we will improve the organization of the introduction and methodology sections so that the motivation and prior work are more clearly separated from the proposed method.

---

> > ### Author Rebuttal · Reviewer_xgZA · 2026-04-04
> >
> > Thank you for the rebuttal . You have addressed my main questions. I keep my original rating.

---

> > > ### Author Response · Authors · 2026-04-04
> > >
> > > Thank you for the positive evaluation and for acknowledging our rebuttal. We appreciate your time and thoughtful feedback, and we are glad the clarifications were helpful.

---

### Official Review · Reviewer_WwLm · 2026-03-12

**Soundness:** 3
**Presentation:** 3
**Significance:** 2
**Originality:** 2
**Overall Recommendation:** 4
**Confidence:** 3

**Summary:**

The paper investigates scalability in federated unsupervised multi-source domain adaptation (UMDA). The proposed method, GALA, replaces costly pairwise domain alignment with inter-group discrepancy (IGD)—i.e., randomly splitting sources into two groups and minimizing the disagreement on unlabeled target data—and introduces temperature-scaled centroid weighting for adaptive source relevance estimation. Experiments on Digit-Five, Office-Caltech10, and the newly proposed Digit-18 benchmark demonstrate the competitive performance of GALA.

**Compliance With Llm Reviewing Policy:**

Affirmed.

**Final Justification:**

I appreciate the authors' detailed responses, which have addressed my concerns well. I would like to increase my score to a Weak Accept. Please ensure these responses are included in the revised version of the paper.

**Key Questions For Authors:**

Building on the first weakness, GALA requires the participation of all sources in every round, whereas FACT only requires one pair. Is there a potential intermediate design that could achieve a better trade-off between performance and computational efficiency?

**Limitations:**

Yes.

**Strengths And Weaknesses:**

**Strengths**

- IGD reduces pairwise alignment complexity from $O(N^2)$ to $O(1)$ per round. This scalability advantage is clearly demonstrated; GALA remains stable as the number of sources grows to 17, while baselines degrade.

- The proposed Digit-18 benchmark provides a meaningful scalability evaluation for the federated domain adaptation community.

- The ablation study in Figure 5 shows that IGD and MDMGB+ are complementary in GALA, supporting the solid methodological design.

**Weaknesses**

- GALA only marginally outperforms FACT in Tables 1 and 2, yet FACT is significantly more efficient than GALA according to Table 4. Although the performance gain of GALA on Digit-18 is substantial, the current experiments do not comprehensively demonstrate a clear advantage of GALA over FACT when both accuracy and efficiency are considered.

- Current experiments are mainly limited to digit-based or small-scale datasets such as Office-Caltech10. Since scalability bottlenecks are a critical real-world consideration, using relatively small or "toy" datasets undermines the significance of this motivation.

- The proposed Digit-18 benchmark is, to some extent, manually curated. It remains unclear whether it accurately represents real-world domain shifts, making the scalability findings difficult to generalize to practical federated settings.

---

> ### Author Rebuttal · Authors · 2026-03-30
>
> Thank you for the thoughtful feedback. We appreciate the reviewer’s concerns regarding the performance–efficiency trade-off, dataset complexity and the insightful question regarding intermediate designs. We respond to all raised weaknesses and questions below.
>
> **(1) Performance vs. efficiency trade-off**
>
> We would like to clarify that GALA’s higher runtime in Table 4 is not caused by full client participation. In our setting, clients train locally in parallel, which is standard in federated learning. FACT only involves one source pair per round, but the remaining clients are effectively idle during that time. As a result, full participation in GALA does not result in a proportional increase in total training time.
>
> The additional overhead in GALA mainly comes from local centroid computation and server-side similarity weighting. This explains why runtime increases with the number of clients. We also note that GALA’s waiting cost is governed by the slowest client among the participating set, but in our experiments this effect is limited because local training times are relatively similar across domains.
>
> Most importantly, we believe this moderate overhead is justified in the many-domain setting, which is the main focus of our work. On Digit-18, GALA provides substantially better accuracy and much more stable training than FACT, while avoiding the severe runtime explosion of KD3A (extrapolating the trend in Table 4 suggests that KD3A would take roughly 2 months **per round** in the 17 source Digit-18 setting). In smaller settings such as Digit-5 or Office-Caltech10, where existing methods already perform strongly, the efficiency advantage of FACT is more pronounced. Our goal, however, is to enable effective federated domain adaptation in the harder many-domain regime.
>
> **(2) Intermediate designs and trade-offs**
>
> GALA naturally supports intermediate designs. To examine this, we conducted additional experiments on Digit-18 by randomly selecting only a fraction of clients to participate in each round (mean and standard deviation are reported over three independent runs).
>
> |Target|MNISTM|SVHNXS|
> |:-|:-|:-|
> |GALA (Full participation)|95.17±0.05|88.47±0.05|
> |GALA (75% participation)|94.73±0.25|88.67±0.29|
> |GALA (50% participation)|94.00±0.33|87.93±0.12|
> |GALA (25% participation)|92.40±0.64|84.20±1.92|
> |FACT|87.03±1.47|80.57±4.90|
>
> These results show that reducing participation yields a gradual trade-off, with only a slight drop in accuracy while preserving the stability of GALA relative to FACT. This indicates that GALA can indeed be adapted to different computation budgets. Nevertheless, reducing participation does not drastically cut total training time (due to the parallelization mentioned above) and the runtime reduction is comparable to decreasing the total number of clients (Table 4).
>
> We further observed that the learned client weights stabilize after several rounds, as shown in Appendix Fig. 10. This suggests a possible extension in which clients with consistently low weights could be excluded in later rounds, which may further improve efficiency. We consider this a promising direction for future work.
>
> **(3) Dataset scale**
>
> Constructing benchmark datasets that capture real-world domain shifts while still enabling controlled, standardized comparison across methods is inherently challenging. To the best of our knowledge, no existing benchmark covers the high-diversity, many-domain setting that we target. We therefore view Digit18 as a first step in this direction. We would like to point out that Digit18 also incorporates datasets such as ARDIS, ARDISII, and TMNIST and thus includes real-world domain shifts in addition to those from Digit5. Furthermore, as shown in Appendix Fig. 7, the benchmark shows substantial cross-domain differences, supporting its usefulness for scalability analysis.
>
> Following the reviewer’s suggestion, we evaluated GALA on DomainNet:
>
> | |Clipart|Infograph|Painting|Quickdraw|Real|Sketch|Avg|
> |-|-|-|-|-|-|-|-|
> |KD3A*|69.7|**21.2**|**58.8**|15.1|**70.4**|57.9|48.85|
> |GALA|**72.0**|20.8|56.4|**20.2**|68.9|**58.4**|**49.45**|
>
> *without data augmentation
>
> For simplicity, we did not use any data augmentation. Although DomainNet has relatively few domains (6) and is not our primary target setting, GALA remains competitive and achieves better average performance than KD3A. These results support that GALA extends beyond digit benchmarks while maintaining its advantages. We will include a full comparison table on DomainNet in the camera-ready manuscript.
>
> ---
>
> We thank the reviewer again for the constructive comments and feedback. We have addressed the raised concerns through additional experiments and clarifications, and we believe these revisions significantly strengthen the manuscript. We would be grateful if the reviewer could reconsider their evaluation in light of the new results.

---

> > ### Author Rebuttal · Reviewer_WwLm · 2026-04-04
> >
> > I appreciate the authors' detailed responses, which have addressed my concerns well. I would like to increase my score to a Weak Accept. Please ensure these responses are included in the revised version of the paper.

---

> > > ### Author Response · Authors · 2026-04-04
> > >
> > > Thank you for your positive acknowledgement. We sincerely appreciate your thoughtful feedback and are glad our rebuttal addressed your concerns. We will incorporate the relevant content into the revised version.

---

### Official Review · Reviewer_yiTW · 2026-03-12

**Soundness:** 2
**Presentation:** 2
**Significance:** 2
**Originality:** 2
**Overall Recommendation:** 2
**Confidence:** 4

**Summary:**

The paper proposes GALA, a federated Unsupervised Multi-Source Domain Adaptation (UMDA) framework designed to address scalability bottlenecks in high-diversity source settings. The authors introduce an Inter-Group Discrepancy (IGD) loss, which randomly splits source predictors into two groups and minimizes their aggregated $l_1$ disagreement on the target data. To handle negative transfer, they utilize MDMGB+, a temperature-scaled, centroid-based similarity weighting scheme. They evaluate GALA on Digit-Five, Office-Caltech10, and a newly proposed 18-source benchmark, Digit-18. While the empirical results on these specific datasets show improvements, critical concerns remain regarding the lack of rigorous theoretical proofs, the reliance on low-complexity datasets to prove "scalability," and ambiguities in the FL topology.

**Compliance With Llm Reviewing Policy:**

Affirmed.

**Ethical Review Concerns:**

Possibly none.

**Final Justification:**

I sincerely thank the authors for providing detailed responses in an attempt to clarify my concerns raised in my review and during the discussion phase, post rebuttal. I also thank the reviewers for acknowledging the weaknesses in their works and hope they will address them in future versions. The paper is below acceptance threshold in its present form and needs better positioning and less ambiguity in its empirical discussions.

1. In my original review, I flagged the lack of statistical significance. Since, the result-margins on Domain-Net are so narrow, I could not comprehend the results meaningfully without any statistical significance. Moreover, it is not clear which models were used for the evaluation in the experiment or how many rounds of FL training it had.

2. I'm glad the reviewers provided the variance comparison between the IDD and IGD. As I can infer from those results I could not see a $1/M$ scaling between the two variances on comparison. This implies that the proof provided in a binary classification case provides a weak lower bound and needs some work to be strengthened and properly formalized.

3. The authors should try to bound the bias theoretically as the number of sources increases to strengthen the paper. For further work, they can provide a formal mapping of reducing variance to generalization.

In view of these, I would like to retain my original rating.

**Key Questions For Authors:**

- (Proof of Variance and Empirical Bias) In Appendix B, you state that IGD is a biased estimator but has lower variance . Please provide a formal mathematical proof for the variance reduction compared to IDD. Furthermore, can you empirically demonstrate that this bias is indeed "small" in practice across your datasets?

- (Scalability to Complex Domains) The inflection point where current methods break is shown only on simple Digit datasets. To truly claim scalability, please provide experiments on DomainNet (e.g., the 10-class or full version) or a multi-source split of CIFAR-100. Do the centroid similarity metrics hold up in high-dimensional feature spaces?

- (Notation in Equations 8 and 9) Equation 7 sums the similarity over all classes $C$. However, Equations 8 and 9 retain the $c$ index in the similarity function $S(r_T^c, r_n^c)$. Could you clarify if the weights are computed per-class or per-domain?

**Limitations:**

No. The authors discuss the computational cost of full source participation , but they fail to address the limitations of their centroid-based similarity metric in high-dimensional visual spaces. Furthermore, they need to explicitly discuss the limits of their "intuition-based" proofs regarding the bias introduced by the IGD estimator.

**Strengths And Weaknesses:**

Strengths:

- The motivation to reduce the variance of pairwise discrepancy estimators (like IDD) by grouping predictors is a logical engineering step.

- The material in the paper is well articulated.

- Addressing multi-source heterogeneity in FL is a highly relevant problem.

- Synthesizing centroid-similarity with a group-wise discrepancy metric is an interesting systems approach.

Weaknesses:

- The claim that IGD has "lower variance" but is a "biased approximation" relies almost entirely on intuition analysis in the appendix. The authors use Jensen's inequality to show a bound, but do not provide a rigorous mathematical proof of the variance reduction, nor do they empirically demonstrate that the bias introduced is negligible in practice. Also, The mathematical formulation of the group-normalized weights $\tilde{w_n}$ introduces a vulnerability to random splits. If a highly heterogeneous setting contains a small set of relevant sources $S_{good}$ and a large set of noisy/irrelevant sources $S_{bad}$, a random split might assign all or most of $S_{bad}$ to $\mathcal{G}_2$.

- Given the marginal improvements on certain standard benchmarks, it is difficult to distinguish true algorithmic gains from statistical noise without proper confidence intervals or significance testing on harder datasets.

- The physical location of the target data is not well-motivated in the text. Algorithm 1 implies the target domain acts as its own distinct client/server node where the IGD loss is minimized , but this topological assumption is not clearly justified in the context of standard FL. Additionally, the notation in Equations 8 and 9 uses $S(r_T^c, r_n^c)$ , leaving the index $c$ hanging, even though Equation 7 already summed over $C$.

- The claims of "scale" and "high-diversity" are entirely bounded by toy datasets (Digits and Office-Caltech10). In the Domain Adaptation community, demonstrating true scalability requires evaluation on complex, high-dimensional datasets like DomainNet or CIFAR-100.

- The novelty is severely limited without a formal theoretical proof of the IGD variance/bias trade-off. Re-using Ben-David's classical bounds as "intuition" does not constitute a novel theoretical contribution.

---

> ### Author Rebuttal · Authors · 2026-03-30
>
> We thank the reviewer for the constructive feedback. We respond to all raised weaknesses and questions below.
>
> **(1) Proof of variance reduction and empirical bias**
>
> We appreciate the opportunity to strengthen the analysis of IGD with a formal proof of variance reduction relative to IDD.
>
> Consider the following setting. Assume $2M$ source predictors $F_1,\dots,F_{2M}$ and a random balanced partition into $G_1,G_2$ with uniform weights $\tilde w_i=1/M$. Assuming binary classification, we decompose the L1 norm as sum of coordinates. Noticing that one coordinate of $F_i(G(x))-F_j(G(x))$ lies in $[0,1]$, write it as $h_a(x)-h_b(x)$, $a,b\in\{i,j\}$. The other coordinate is $(1-h_a(x))-(1-h_b(x))=h_b(x)-h_a(x)\in[-1,0]$, since $F$ outputs a probability vector. Also let $\bar h_G(x)=1/|G|\sum_{k\in G}h_k(x)$ and $\bar h(x)$ the average over all $h_k(x)$.
>
> We aim to compare:
>
> $Var(IDD)=V_{a,b}[E_x [2(h_a(x)-h_b(x))]] \approx 8 V_k[ E_x[h_k(x)]]$
>
> $Var(IGD)=V_{G_1}[E_x[2(\bar h_{G_1}(x)-\bar h_{G_2}(x))]]=V_{G_1}[E_x[2(2\bar h_{G_1}(x)-2\bar h(x))]].$
>
> Then,
>
> \begin{align}
> Var(IGD)&=16V_{G_1}[E_x[\bar h_{G_1}(x)]] \quad(E_x\bar h(x)\text{ is constant in }G_1))\\\\
> &=16V_{G_1}[\frac1M\sum_{i\in G_1}E_x[h_i(x)]]\\\\
> &=16\cdot\frac{2M-M}{2M-1}\frac{V_k[E_x[h_k(x)]]}{M}\quad\text{(sample mean w/o replacement)}\\\\
> &\approx \frac8MV_k[E_x[h_k(x)]]\\\\
> &\approx \frac1M\text{variance of IDD}.
> \end{align}
>
> This $1/M$ scaling behavior was verified under a simulation experiment where $h_i(x) \sim Unif[A_i, B_i]$.  Empirically, results on Digit-18 (Table 3) and increasing-source experiments (Fig. 2) show significantly improved stability and performance over IDD (FACT), indicating that the variance reduction is beneficial in practice.
>
> **(2) Scalability and evaluation on more complex datasets**
>
> We clarify that our scalability claim targets the number of distinct source domains. To the best of our knowledge, no existing benchmark captures the high-diversity, many-domain regime we study. We therefore view Digit-18 as a first step in this direction.
>
> Following the reviewer’s suggestion, we evaluated GALA on DomainNet:
>
> | |Clipart|Infograph|Painting|Quickdraw|Real|Sketch|Avg|
> |-|-|-|-|-|-|-|-|
> |KD3A*|69.7|**21.2**|**58.8**|15.1|**70.4**|57.9|48.85|
> |GALA|**72.0**|20.8|56.4|**20.2**|68.9|**58.4**|**49.45**|
>
> *without data augmentation
>
> For simplicity, we did not use any data augmentation. Although DomainNet has relatively few domains (6) and is not our primary target setting, GALA remains competitive and achieves better average performance than KD3A. These results support that GALA extends beyond digit benchmarks while maintaining its advantages. We will include a full comparison table on DomainNet in the camera-ready manuscript.
>
> **(3) Random split vulnerability**
>
> We would like to clarify that the two groups are re-sampled randomly at the beginning of every training round, rather than fixed throughout training, so no single partition persists across rounds. Empirically, Fig. 3 (third subplot) captures the exact heterogeneous setting described by the reviewer. On SVHNXS in Digit-18, SYNXS is the only dominant source, and GALA assigns it a very large weight (>0.55), while the remaining 16 sources together receive only about 0.45. In this setting, FACT is substantially more vulnerable because IDD samples only one random source pair per round and is therefore more sensitive when only a small subset of sources is relevant. By contrast, GALA remains robust throughout training, and Table 3 shows that it outperforms FACT by a large margin and even exceeds the Oracle performance.
>
> **(4) significance testing**
>
> Our results already report mean ± std over five independent runs. The gains on standard benchmarks are modest but consistent across datasets and runs, indicating that they are not caused by outliers. As these benchmarks are already near saturation, large absolute improvements are unlikely. Our main contribution is the high-source regime, where the gains are substantially larger and the robustness improvements are clear. We will also add confidence intervals in an extended appendix table to further address this concern.
>
> **(5) Notation in Eq. 8 & 9**
>
> Thank you for pointing out the ambiguity. You are correct, this is a notation issue. We will revise Eqs. 8 and 9 to replace $S(r_T^c, r_j^c)$ with $S(r_T, r_j)$.
>
> **(6) Target location & FL topology**
>
> We agree this can be clarified. The target node is an optimization abstraction, not a deployment assumption. As shown in Algorithm 1, sources perform local updates, the server aggregates relevance-weighted models, and the target updates the shared extractor via IGD. We will revise the text to make this more clear to the readers.
>
> ---
>
> We thank the reviewer again for the insightful feedback, and we believe the added theoretical analysis and new experimental results directly address all the concerns raised. We would be grateful if the reviewer could reconsider their current score.

---

> > ### Author Rebuttal · Reviewer_yiTW · 2026-04-04
> >
> > I thank the reviewers for their efforts in clarifying some of my concerns. I appreciate the commitment to fixing the indexing in Equations 8 and 9. These presentation issues are now resolved.
> >
> > However, there are several remaining concerns such as the proof relying on overly simplistic assumptions. The authors have included a table containing some results on the DomainNet dataset. The numbers reported are not very convincing. But more importantly, the authors have failed to provide any meaningful interpretation of/insight into these results. Furthermore, the authors haven’t addressed another major concern regarding the inherent bias of the IGD estimator as they haven’t provided the requested empirical evidence to demonstrate that this bias is negligible across their datasets.
> >
> > I would request the authors to provide clarifications on the following points -
> >
> > On the variance: Since, the paper uses non-uniform, temperature scaled weights I am not sure how well the uniform weights assumption holds in the proof. How about, you assume them to lie on a probability simplex (to try and capture their non-uniformity)? The derivation also treats predictors as independent samples; however, within the GALA framework, all predictors $F_n$ are fine-tuned on a $G’$. This coupling introduces inter-predictor correlations, so how valid is the independence assumption?
> >
> > On the Domain-Net experiment: Reporting the numbers honestly is great. Please try to provide insights on why we may be seeing a drop in half of the settings. Does the centroid-based similarity metric become less reliable in the higher-dimensional feature spaces of DomainNet compared to the Digit datasets? Feel free to point out limitations of your approach and under what conditions may the best results be obtained.

---

> > > ### Author Response · Authors · 2026-04-07
> > >
> > > Thank you for these careful follow up questions. We appreciate the chance to clarify the remaining concerns.
> > >
> > > **Q1.1 Independence**
> > >
> > > **A1.1** There is no independence assumed between $h_a$ and $h_b$. The derivation relies on standard combinatorial results for selecting (a,b) and $G_1$, and assumes only:
> > >
> > > 1. $x \sim D_T$ is independent of $G_1$ and (a,b), consistent with IDD and IGD.
> > > 2. $G_1$ is a size-M subset of the 2M sources sampled w.o replacement, consistent with IGD.
> > > 3. (a,b) is drawn uniformly w.o replacement from the 2M sources, consistent with IDD.
> > >
> > > We also found that the $\approx$ in the calculation of Var(IDD) is actually an equality. Let $\mu_k = E_x[h_k(x)]$. Then
> > >
> > > $$V_{a,b}[\mu_a-\mu_b]=V_{a,b}[\mu_a]+V_{a,b}[\mu_b]-2Cov_{a,b}(\mu_a,\mu_b)$$
> > >
> > > Let $\sigma^2 = \frac{1}{2M}\sum_k(\mu_k - \bar\mu)^2$ and $S^2 = \frac{1}{2M-1}\sum_k(\mu_k - \bar\mu)^2$. Since a is drawn uniformly from 2M sources, $V_a[\mu_a]=\sigma^2$. For two draws w.o replacement from a population of size 2M, $Cov_{a,b}(\mu_a,\mu_b)=-\frac{\sigma^2}{2M-1}$. Therefore, $V_{a,b}[\mu_a - \mu_b] = 2\sigma^2 + \frac{2\sigma^2}{2M-1} = 2S^2.$ Combined with $Var(IGD)=\frac{16S^2}{2M-1}$ we get
> > >
> > > $$\frac{Var(IGD)}{Var(IDD)} = \frac{2}{2M-1}\approx\frac{1}{M}.$$
> > >
> > > No independence assumption is needed because $\mu_k$ is deterministic for each trained predictor. Correlations from shared fine-tuning do not affect the randomness from (a,b) or $G_1$.
> > >
> > > **Q1.2 Non-uniform Weights**
> > >
> > > **A1.2** $Var(IGD) \le Var(IDD)$ does **not** hold for any arbitrary non-uniform weights. Counterexamples can be easily constructed, e.g. when large weights are assigned to sources with conflicting predictions.
> > >
> > > In GALA, however, weights are data-driven and larger weights are assigned to sources similar to the target, so such adversarial configurations are unlikely in practice. We further support this in A2 with empirical evidence that our centroid-based similarity reliably identifies relevant sources, even in high-dimensional settings such as DomainNet.
> > >
> > > We emphasize that the uniform weight assumption in Appendix is not a claim about how GALA works in practice, but a useful theoretical anchor that gives an exact characterization of the variance reduction mechanism. In A1.3, we empirically compute the variance reduction ratio using the non-uniform weights learned by GALA.
> > >
> > > **Q1.3 Bias of IGD**
> > >
> > > **A1.3** We agree that our earlier claim was too strong. A more accurate claim is that in practice, the effect of bias on classification performance seems negligible.
> > >
> > > Empirically, if the downward bias were harmful, GALA should underperform FACT on easy target domains where both methods converge. Instead, GALA consistently improves stability and accuracy, especially as the number of sources grows, with no evidence of underfitting expected from a strongly biased estimator. Following the reviewer’s suggestion, we also include an explicit comparison of $L_{full}$, IGD, IDD in (https://anonymous.4open.science/r/GALA_Rebuttal-468B/IGD_empirical_analysis.pdf), which confirms that IGD with non-uniform weights learned by GALA is a biased but low-variance estimator as hypothesized.
> > >
> > > Lastly, the convergence analysis (Fig. 2 in supplementary repo) shows that IGD closely tracks $L_{full}$; thus minimizing $L_{IGD}$ also effectively minimizes $L_{full}$ without evaluating all source combinations. Overall, these results suggest that although the bias exists, its impact on classification accuracy is negligible and variance reduction is observed in practice.
> > >
> > > **Q2 DomainNet Insights and Metric Reliability**
> > >
> > > **A2** Our main claim is scalability with respect to the number of heterogeneous sources in distributed UMDA. The increasing-source experiments and Digit18 show that GALA is more robust, while remaining efficient. The additional results on Digit-Five, Office-Caltech10, and DomainNet are intended to show that GALA is competitive on standard benchmarks, rather than to claim significant improvements on all targets.
> > >
> > > On DomainNet, even with default parameters, GALA achieves higher average accuracy than KD3A (the current SOTA), with a notable gain on quickdraw, the most challenging target. These results are consistent with our findings on other low-diversity benchmarks, where GALA similarly improves on a subset of targets, and therefore do not indicate a limitation of the method in harder, higher-dimensional settings.
> > >
> > > To examine the centroid-based similarity metric, we compare GALA’s source weights with KD3A’s in (https://anonymous.4open.science/r/GALA_Rebuttal-468B/domain_similarity_analysis.pdf). The two are broadly consistent, suggesting that the metric remains meaningful on DomainNet. The main exception is quickdraw, where the methods differ, however, GALA performs better there in our rebuttal results (15.1 vs. 20.2), which supports the strength of our similarity metric.

---

### Decision · Program_Chairs · 2026-04-30

**Decision:**

Accept (regular)

**Comment:**

The original concerns raised by the reviewers mainly concern the experimental results and some clarifications of the theoretical results. After the rebuttal, most of the concerns have been addressed. As one reviewer pointed out, the current theoretical results are not very strong and can be further improved.

Overall, this is a borderline paper with some merit. Thus, I recommend a weak acceptance.